# Block-local learning with probabilistic latent representations

## Abstract

The ubiquitous backpropagation algorithm requires sequential updates across blocks of a network, introducing a locking problem. Moreover, backpropagation relies on the transpose of weight matrices to calculate updates, introducing a weight transport problem across blocks. Both these issues prevent efficient parallelisation and horizontal scaling of models across devices. We propose a new method that introduces a twin network that propagates information backwards from the targets to the input to provide auxiliary local losses. Forward and backward propagation can work in parallel and with different sets of weights, addressing the problems of weight transport and locking. Our approach derives from a statistical interpretation of end-to-end training which treats activations of network layers as parameters of probability distributions. The resulting learning framework uses these parameters locally to assess the matching between forward and backward information. Error backpropagation is then performed locally within each block, leading to "block-local" learning. Several previously proposed alternatives to error backpropagation emerge as special cases of our model. We present results on various tasks and architectures, including transformers, demonstrating state-of-the-art performance using block-local learning. These results provide a new principled framework to train very large networks in a distributed setting and can also be applied in neuromorphic systems.

## 1 Introduction

Recent developments in machine learning have seen deep neural network architectures scaling to billions of parameters [Touvron et al., 2023, Brown et al., 2020]. This development has boosted the capabilities of these models to unprecedented levels but simultaneously pushed the computing hardware on which large network models are running to its limits. It is therefore becoming increasingly important to distribute learning algorithms over a large number of independent compute nodes. However, today's machine learning algorithms are ill-suited for distributed computing. The error backpropagation (backprop) algorithm requires an alternation of inter-depended forward and backward phases, introducing a locking problem (the two phases have to wait for each other) [Jaderberg et al., 2016a]. Furthermore, the two phases rely on the same weight matrices to calculate updates, introducing a weight transport problem across blocks [Grossberg, 1987, Lillicrap et al., 2014a]. These two issues make efficient parallelisation and horizontal scaling of large machine learning models across compute nodes extremely difficult.

We propose a new method to address these problems by distributing a globally defined optimisation algorithm across a large network of nodes that use only local learning. Our approach uses a message-passing approach that uses results from probabilistic models and communicates uncertainty messages forward and backwards between compute nodes in parallel. To do so, we augment a network

architecture with a twin network that propagates information backwards from the targets to the input to provide uncertainty measures and auxiliary targets for local losses. Forward and backward messages comprise information about extracted features and feature uncertainties and are matched against each other using local probabilistic losses. Importantly, forward and backward propagation can work in parallel, reducing the locking problem. Inside each block, conventional error backpropagation is performed locally ("block-local"). These local updates can be used in the forward network and its backward twin for adapting parameters during training. The developed theoretical learning provides a new principled method to distribute very large networks over multiple compute nodes. The solutions emerging from this framework show striking similarities to earlier models that used random feedback weights as local targets [Lillicrap et al., 2020, Frenkel et al., 2021] but also provide a principled way to train these feedback weights.

In summary, the contribution of this paper is threefold:

1. We provide a theoretical framework on how interpreting the representations of deep neural networks as probability distributions provides a principled approach for block-local training of these networks. This can be used to distribute learning and inference over many interacting neural network blocks for various neural network architectures.

2. We demonstrate an instance of this probabilistic learning model on several benchmark classification tasks, where classifiers are split into multiple blocks and trained without end-to-end gradient computation.

3. We demonstrate how this framework can be used to allow deep networks to produce uncertainty estimates over their predictions. This principle is showcased on an autoencoder network that automatically predicts uncertainties alongside pixel intensity values after training.

## 2    Related work

A number of methods for using local learning in DNNs had been introduced previously. Lomnitz et al. [2022] introduced Target Projection Stochastic Gradient Descent (tpSGD), which uses layer-wise SGD and local targets generated via random projections of the labels, but does not adapt the backward weights. LocoProp [Amid et al., 2022] uses a layer-wise loss that consists of a target term and a regularizer, which is used however to enable 2nd order learning and does not focus on distributing the gradient optimization. Jimenez Rezende et al. [2016] used a generative model and a KL-loss for local unsupervised learning of 3D structures.

Some previous methods are based on probabilistic or energy-based cost functions and use a contrastive approach with positive and negative data samples. Contrastive learning Chen et al. [2020], Oord et al. [2019] can be used to construct block-local losses Xiong et al. [2020], Illing et al. [2021]. Equilibrium propagation replaces target clamping with a target nudging phase [Scellier and Bengio, 2017]. Another interesting contrastive approach was recently introduced [Hinton, 2022, Ororbia and Mali, 2023, Zhao et al., 2023]. However, it needs task-specific negative examples. [Han et al., 2018] uses a local predictive loss to improve recurrent networks' performance. In contrast to these methods, our approach does not need separate positive and negative data samples and focuses on block-local learning.

Feedback alignment [Lillicrap et al., 2020, Sanfiz and Akrout, 2021] uses random projections to propagate gradient information backwards. Jaderberg et al. [2016b] used pseudo-reward functions which are optimized simultaneously by reinforcement learning to improve performance. Random feedback alignment [Amid et al., 2022, Refinetti et al., 2021] and related approaches [Clark et al., 2021, Nøkland, 2016, Launay et al., 2020], use fixed random feedback weights to back-propagate errors. [Jaderberg et al., 2017] used decoupled synthetic gradients for local training. Target propagation demonstrates non-trivial performance with random projections for target labels instead of errors [Frenkel et al., 2021]. In contrast to these methods, we provide a principled way to adapt feedback weights.

Other methods [Belilovsky et al., 2019, Löwe et al., 2019] used greedy local, block- or layer-wise optimization. Notably, Nøkland and Eidnes [2019] achieved good results by combining a matching and a local cross-entropy loss. [Siddiqui et al., 2023] recently used block-local learning based on a cross-correlation metric over feature embeddings [Zbontar et al., 2021], demonstrating promising

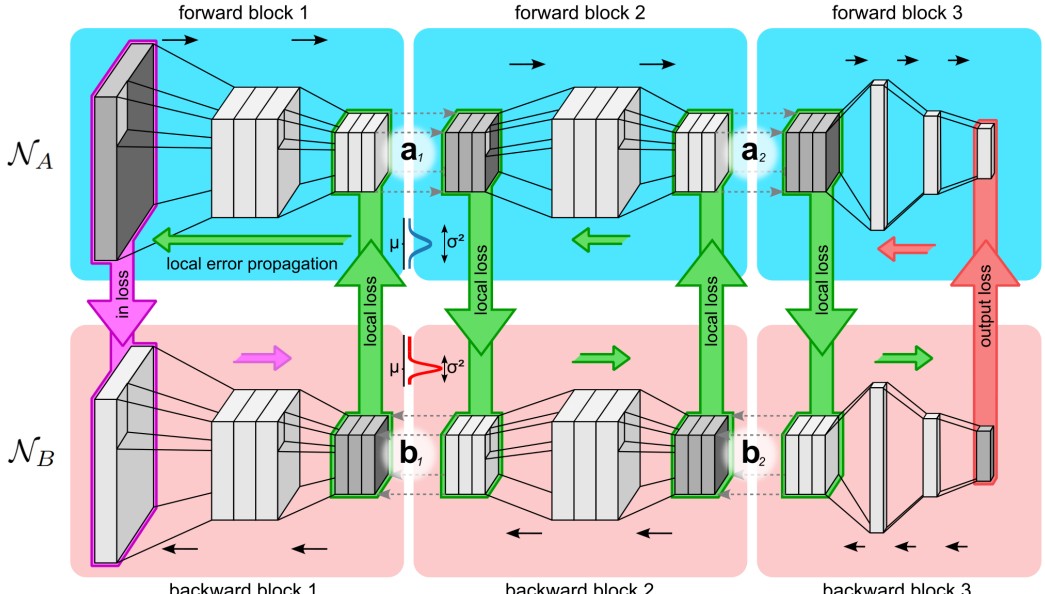

Figure 1: Illustration of use of block-local representations as learning signals on intermediate network layers. A deep neural network architecture $\mathcal{N}_A$ is split into multiple blocks (forward blocks) and trained on an auxiliary local loss. Targets for local losses are provided by a twin backward network $\mathcal{N}_B$.

performance. [Wu et al., 2021] used greedy layer-wise optimization of hierarchical autoencoders for video prediction. [Wu et al., 2022] used an encoder-decoder stage for pretraining. In contrast to these methods, we do not rely solely on local greedy optimization but provide a principled way to combine local losses with feedback information without locking and weight transport across blocks.

## 3 A probabilistic formulation of distributed learning

At a high level, our method interprets the activations of a neural network as the parameters of probability distributions of latent variables. We use these intermediate representations at each block to derive block local losses. These latent variables over multiple blocks implicitly define a Markov chain, which allows us to tractably minimize the block's local loss. We show that the derived block local losses and the resulting block local learning (BLL) are a general form of various existing local losses and provide an upper bound to a global loss.

### 3.1 Using latent representations to construct probabilistic block-local losses

Learning in deep neural networks can be formulated probabilistically [Ghahramani, 2015] in terms of maximum likelihood, i.e. the problem is to minimize the negative log-likelihood $\mathcal{L} = -\log p(\mathbf{x}, \mathbf{y}) = -\log p(\mathbf{y} \mid \mathbf{x}) - \log p(\mathbf{x})$ with respect to the network parameters $\boldsymbol{\theta}$. For many practical cases where we may not be interested in the prior distribution $p(\mathbf{x})$, we would like to directly minimize $\mathcal{L} = -\log p(\mathbf{y} \mid \mathbf{x})$.

This probabilistic interpretation of deep learning can be used to define block-local losses and distribute the learning over multiple blocks of networks by introducing intermediate latent representations. The idea is illustrated in Fig. 1. A neural network that computes the distribution $\log p(\mathbf{y} \mid \mathbf{x})$ takes $\mathbf{x}$ as input and outputs the statistical parameters to the conditional distribution. The deep neural network is split at an intermediate layer $k$ (in Fig. 1 we used $k \in (1, 2)$) and end-to-end estimation of the gradient is replaced by two estimators that optimize the sub-networks $\mathbf{x} \to \mathbf{z}_k$ and $\mathbf{z}_k \to \mathbf{y}$ separately. To do this, consider the gradient of the log-likelihood loss function

$$-\frac{\partial}{\partial \theta}\mathcal{L} = \frac{\partial}{\partial \theta} \log p(\mathbf{y} \mid \mathbf{x}) \ . \tag{1}$$

For any deep network, it is possible to choose any intermediate activation at layer $k$ as latent representations $\mathbf{z}_k$, such that $\log p\left(\mathbf{y} \mid \mathbf{x}\right) = \left\langle p\left(\mathbf{y} \mid \mathbf{z}_k\right) p\left(\mathbf{z}_k \mid \mathbf{x}\right)\right\rangle_{p\left(\mathbf{z}_k \mid \mathbf{x}, \mathbf{y}\right)}$, where $\left\langle \, \right\rangle_p$ denotes expectation with respect to $p$. Therefore, the representations of $\mathbf{y}$ depend on $\mathbf{x}$ only through $\mathbf{z}_k$ as expected for a feed-forward network. Using this conditional independence property, the log-likelihood (1) expands to

$$-\frac{\partial}{\partial \theta}\mathcal{L} \;=\; \frac{\partial}{\partial \theta}\log p\left(\mathbf{y} \mid \mathbf{x}\right) \;=\; \left\langle \frac{\partial}{\partial \theta}\log p\left(\mathbf{y} \mid \mathbf{z}_k\right) + \frac{\partial}{\partial \theta}\log p\left(\mathbf{z}_k \mid \mathbf{x}\right)\right\rangle_{p\left(\mathbf{z}_k \mid \mathbf{x}, \mathbf{y}\right)} . \qquad (2)$$

This well-known result is the foundation of the Expectation-Maximization (EM) algorithm [Dempster et al., 1977]. Computing the marginal with respect to $p\left(\mathbf{z}_k \mid \mathbf{x}, \mathbf{y}\right)$ corresponds to the E-step and calculating the gradients corresponds to the M-step. The sum inside the expectation separates the gradient estimators into two parts: $\mathbf{x} \to \mathbf{z}_k$ and $\mathbf{z}_k \to \mathbf{y}$.

However, the E-step is impractical to compute for most interesting applications because of the combinatorial explosion in the state space of $\mathbf{z}_k$. To get around this, we use a variational lower bound to EM, based on the ELBO loss $\mathcal{L}_V \;=\; -\log p\left(\mathbf{y} \mid \mathbf{x}\right) + \mathcal{D}_{KL}\left(q \mid p\right)$ [Mnih and Gregor, 2014] and demonstrate that this yields a practical solution to split gradients in a similar fashion to Eq. (2). In the next section, we describe how we construct the variational distribution $q$.

## 3.2 Auxiliary latent representations

As described earlier, the output of any layer of a DNN can be interpreted as parameters to a distribution over latent random variable $\mathbf{z}_k$. The sequence of blocks across a network therefore implicitly defines a Markov chain $\mathbf{x} \to \mathbf{z}_1 \to \mathbf{z}_2 \to \ldots$ (see Fig. 2A). This probabilistic interpretation of hidden layer activity is valid under relatively mild assumptions, studied in more detail in the Supplement. It is important to note that the network at no point produces samples from the implicit random variables $\mathbf{z}_k$, but they are introduced here only to conceptualize the mathematical framework. Instead the network outputs the parameters to $\alpha_k(\mathbf{z}_k)$ which is the probability distribution over $\mathbf{z}_k$ (e.g. means and variances if $\alpha_k$ is Gaussian). The network thus translates $\alpha_{k-1} \to \alpha_k \to \ldots$ by outputting the statistical parameters of the conditional distribution $\alpha_k(z_k)$ and taking $\alpha_k(z_{k-1})$ parameters as input. More precisely, the network implicitly computes a marginal distribution

$$\alpha_k\left(\mathbf{z}_k\right) \;=\; p\left(\mathbf{z}_k \mid \mathbf{x}\right) \;=\; \left\langle p_k\left(\mathbf{z}_k \mid \mathbf{z}_{k-1}\right)\right\rangle_{p\left(\mathbf{z}_{k-1} \mid \mathbf{x}\right)} \;=\; \left\langle p_k\left(\mathbf{z}_k \mid \mathbf{z}_{k-1}\right)\right\rangle_{\alpha_{k-1}\left(\mathbf{z}_{k-1}\right)} , \qquad (3)$$

where $\left\langle \, \right\rangle_p$ denotes expectation with respect to the probability distribution $p$. Consequently, the network realizes a conditional probability distribution $p\left(\mathbf{y} \mid \mathbf{x}\right)$ (where $\mathbf{x}$ and $\mathbf{y}$ are network inputs and outputs, respectively). And by the universal approximator property of deep neural networks, an accurate representation of this distribution can be learnt in the network weights through error back-propagation (as demonstrated for the example in Fig. 2). Eq. (3) is an instance of the belief propagation algorithm to efficiently compute conditional probability distributions.

To construct the variational distribution $q$ we introduce the backward network $\mathcal{N}_B$ that propagates messages $\beta_k$ backwards according to Eq. 4 (see Fig. 1 for an illustration). Inference over the posterior distribution $p\left(\mathbf{z}_k \mid \mathbf{x}, \mathbf{y}\right)$ for any latent variable $\mathbf{z}_k$ can be made using the belief propagation algorithm, propagating messages $\alpha_k\left(\mathbf{z}_k\right)$ forward through the network using Eq. (3). In addition messages $\beta_k\left(\mathbf{z}_k\right)$ need to be propagated backward according to

$$\beta_k\left(\mathbf{z}_k\right) \;=\; p\left(\mathbf{y} \mid \mathbf{z}_k\right) \;=\; \left\langle p\left(\mathbf{y} \mid \mathbf{z}_{k+1}\right)\right\rangle_{p_k\left(\mathbf{z}_{k+1} \mid \mathbf{z}_k\right)} \;=\; \left\langle \beta_{k+1}\left(\mathbf{z}_{k+1}\right)\right\rangle_{p_k\left(\mathbf{z}_{k+1} \mid \mathbf{z}_k\right)} , \qquad (4)$$

such that the posterior $p\left(\mathbf{z}_k \mid \mathbf{x}, \mathbf{y}\right)$ can be computed up to normalization

$$\rho_k\left(\mathbf{z}_k\right) \;=\; p\left(\mathbf{z}_k \mid \mathbf{x}, \mathbf{y}\right) \quad \propto \quad p\left(\mathbf{z}_k \mid \mathbf{x}\right) p\left(\mathbf{y} \mid \mathbf{z}_k\right) \;=\; \alpha_k\left(\mathbf{z}_k\right) \beta_k\left(\mathbf{z}_k\right) . \qquad (5)$$

We make use of the fact that, through Eq. (3), the parameters of a probability distribution $p\left(\mathbf{z}_k \mid \mathbf{x}\right)$ are a function of the parameters to $p\left(\mathbf{z}_i \mid \mathbf{x}\right)$, for $0 < i < k$, e.g. if $\alpha$ is assumed to be Gaussian we have $\left(\mu\left(\alpha_k\right), \sigma^2\left(\alpha_k\right)\right) = f\left(\mu\left(\alpha_i\right), \sigma^2\left(\alpha_i\right)\right)$, where $\mu\left(.\right)$ and $\sigma^2\left(.\right)$ are the mean and variance of the distribution respectively. Thus, if a network outputs $\left(\mu\left(\alpha_i\right), \sigma^2\left(\alpha_i\right)\right)$ on layer $i$ and $\left(\mu\left(\alpha_k\right), \sigma^2\left(\alpha_k\right)\right)$ on layer $k$, a suitable probabilistic loss function will allow the network to learn

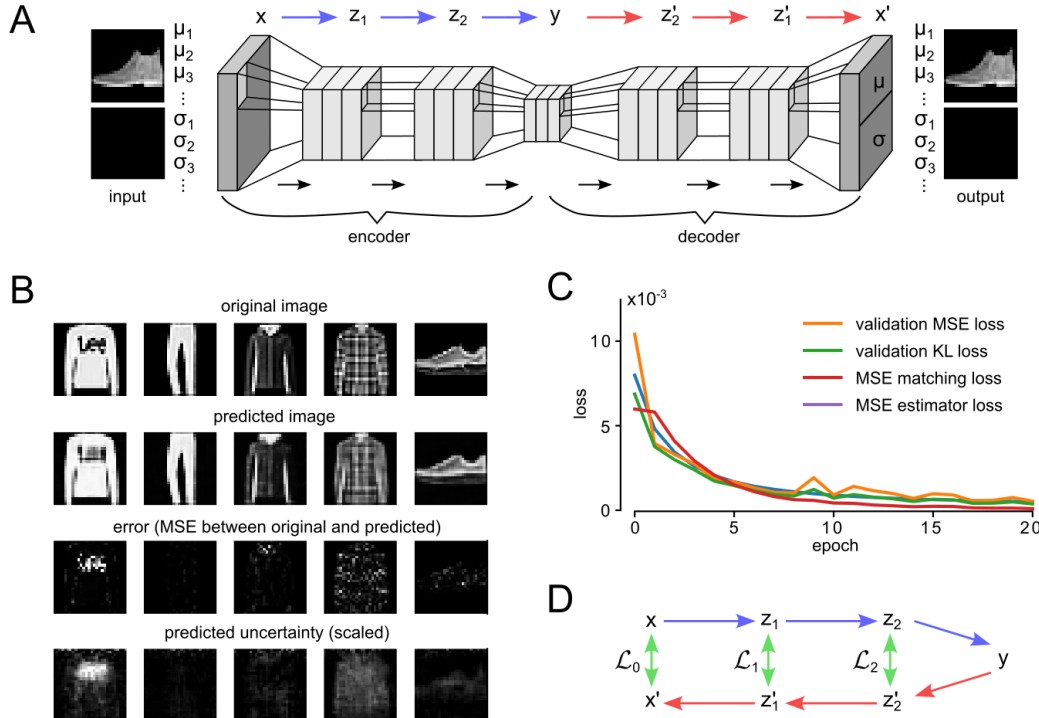

Figure 2: Zero shot learning of predicted uncertainties. **A:** Gaussian convolutional autoencoder network. Variance inputs and outputs are set to a constant during the whole training process. The network implements an implicit Markov chain. **B:** Example images showing self-prediction of uncertainties. **C:** Uncertainty mismatch metrics throughout learning. **D:** The network in (A) can be 'folded' to provide targets for local losses $\mathcal{L}_0, \mathcal{L}_1, \ldots$

$f$ from examples. Therefore, the conditional distributions $p_k(\mathbf{z}_k \mid \mathbf{z}_{k-1})$ and the expectation in Eq. (3) are only implicitly encoded in the network weights. We will study the exponential family of probability distributions for which this observation can be formalized more thoroughly.

**Exponential family distributions:** To derive concrete losses and update rules for the forward and backward networks, we assume that $\alpha_k$ are from the exponential family (EF) of probability distributions, given by

$$\alpha_k(\mathbf{z}_k) = \prod_j \alpha_{kj}(z_{kj}) = \prod_j h(z_{kj}) \exp\left(T(z_{kj})\phi_{kj} - A(\phi_{kj})\right) , \tag{6}$$

with base measure $h$, sufficient statistics $T$, log-partition function $A$, and natural parameters $\phi_{kj}$. This rich class contains the most common distributions, such as Gaussian, Poisson or Bernoulli, as special cases. For the example of a Bernoulli random variable we have $z_{kj} \in \{0, 1\}$, $T(z_{kj}) = z_{kj}$ and $A(\phi_{kj}) = \log\left(1 + e^{\phi_{kj}}\right)$ [Koller and Friedman, 2009]. A network directly implements an EF distribution if the activations $a_{kj}$ encode the natural parameters, $a_{kj} = \phi_{kj}$. Using this result, a feed-forward DNN $\mathcal{N}_A : \mathbf{x} \to \mathbf{y}$, can be split into $N$ blocks by introducing implicit latent variables $\mathbf{z}_k : \mathbf{x} \to \mathbf{z}_k \to \mathbf{y}$, and generating the respective natural parameters. In principle, blocks can be separated after any arbitrary layer, but some splits may turn out more natural for a particular network architecture.

Conveniently, if both $\alpha_{kj}$ and $\beta_{kj}$ are members of the EF with natural parameters $a_{kj}$ and $b_{kj}$, then $\rho_{kj}$ is also EF with parameters $a_{kj} + b_{kj}$. We will use this property to deconstruct a single global loss into multiple block-local losses.

## 3.3 Illustrative example: forward-backward networks as an autoencoder

Probability representations in DNNs are useful since they provide a principled way to represent uncertainties in the network. Before we establish our main result to show how a DNN can be deconstructed into local blocks, we first demonstrate how representations of Bayesian uncertainty can emerge in DNNs by using appropriate probabilistic losses. We consider the autoencoder network illustrated in Fig. 2A and use it to learn representations for the Fashion-MNIST dataset [Xiao et al., 2017]. The CNN comprises a bottleneck layer $\mathbf{y}$ that implicitly splits the architecture into a decoder and encoder part (Fig. 2A). It is well known that such a network is able to learn compact representations and features that allow it to reconstruct the gray scale pixel intensities of a given input [Kingma and Welling, 2013]. Here we demonstrate that autoencoders are also able to learn representations of uncertainties, i.e. to automatically output high uncertainties for pixel values that are poorly represented in the learnt features.

To show this, we augmented the pixel representations on the inputs and outputs with additional channels that represented the logarithms of the variances of a Gaussian distribution (see Supplement for details). The input and outputs now represent the parameters of probability distributions, where the variances are proxies for the uncertainties. An appropriate loss function for this architecture is one that measures the distance between probability distributions. We used the Kullback-Leibler (KL) divergence between Gaussian distributions. This augmentation to conventional deep auto-encoders requires us to also provide uncertainty values for training data samples. Since the Fashion-MNIST dataset does not contain this information, we set the variances of pixels for all training samples to the same small constant values, reflecting high confidence (low variance) in the training set. Thus, during training, the network has only seen the same constant inputs (and outputs) for the variance channels.

Fig. 2B shows representative sample outputs for the test dataset after training. As expected, the network is able to represent the means of gray scale values in the dataset well and generalize to new images. Interestingly, the network also learned meaningful representations of the variances. Although the network has only seen constant values for the variances during training, it is able to infer information about its own uncertainty during testing. The true MSE errors between inputs and predictions qualitatively match the pixel-level variance predictions across a wide variety of inputs. For example, the network poorly represents the logo on the shirt (leftmost example) and predicts high variance in the output for these pixels. Other samples like the trousers (second from left) that are well represented correctly predict low variance. To further quantify this result, we developed additional metrics that measure the mismatch between estimated and true prediction errors (Fig. 2C, see Supplement for details). These metrics consistently decrease throughout training even though they were not directly minimized. These results suggest that DNNs are able to represent uncertainties well enough that they show zero-shot generalizations to unseen data from very limited training data.

## 3.4 Modularized learning using local variational losses

The autoencoder example described in Section 3.3 shows that DNNs can represent probability distributions well in principle, and also provides an idea of how probabilistic losses could be constructed locally at any layer. By 'folding' the network along the bottleneck layer $y$ we are able to construct a sequence of pairs of auxiliary targets $(\mathbf{z}_1, \mathbf{z}_1'), (\mathbf{z}_2, \mathbf{z}_2'), \ldots$ (see Fig. 2D). Finally, by introducing suitable loss functions $\mathcal{L}_0, \mathcal{L}_1, \ldots$, the mismatch between the encoder and decoder parts of the network can be minimized on a per-layer basis.

The forward and backward networks $\mathcal{N}_A$ and $\mathcal{N}_B$ can be used to construct local loss functions $\mathcal{L}_V^{(k)}$ at blocks $k$. In the Supplement, we show in detail that minimizing $\mathcal{L}_V^{(k)}$ locally and in parallel optimizes a lower bound to the log-likelihood loss $\mathcal{L}$ (Eq. 1), without propagating gradients end-to-end. To arrive at this result, we take the forward $\alpha_k$ and posterior messages $\rho_k$ to be given by EF distributions with natural parameters $\phi_{kj}$ and $\gamma_{kj}$. Using this we show in the Supplement that the local loss can be optimized using the modularized gradient estimator

$$-\frac{\partial}{\partial\theta}\mathcal{L}_V^{(k)} = \sum_j \underbrace{\left(\mu\left(\rho_{kj}\right) - \mu\left(\alpha_{kj}\right)\right)}_{forward\ weight} \frac{\partial}{\partial\theta}\phi_{kj} + \underbrace{\sigma^2\left(\rho_{kj}\right)\left(\phi_{kj} - \gamma_{kj}\right)}_{posterior\ weight} \frac{\partial}{\partial\theta}\gamma_{kj}, \qquad (7)$$

where $\mu(\cdot)$ and $\sigma^2(\cdot)$ are means and variances of EF distribution. Note that the gradients of the natural parameters $\phi_{kj}$ and $\gamma_{kj}$ are computed independently and modulated by the *forward* and *posterior weight*, respectively.

The result in Eq. (7) holds for general EF distributions. For the special case of Bernoulli random variables we get

$$-\frac{\partial}{\partial\theta}\mathcal{L}_V^{(k)} \;=\; \sum_{k,j}\left(\rho_{kj}-\alpha_{kj}\right)\frac{\partial}{\partial\theta}a_{kj} \;-\; \rho_{kj}\left(1-\rho_{kj}\right)b_{kj}\left(\frac{\partial}{\partial\theta}a_{kj}+\frac{\partial}{\partial\theta}b_{kj}\right)\;,\qquad(8)$$

where $a_{kj}=f_j(\mathbf{a}_{k-1})$ and $b_{kj}=g_j(\mathbf{b}_{k+1})$, are the outputs of the forward and backward network at block $k$,

$$\rho_{kj}=S\left(a_{kj}+m\,b_{kj}\right)\quad\text{and}\quad\alpha_{kj}=S\left(a_{kj}\right)\;,\qquad(9)$$

where $m$ is a mixing parameter described below and $S(x)=1/1+e^{-x}$ is the sigmoid/logistic function.

The Bernoulli solution in Eq. (8) is convenient because it is a single parameter distribution (mean and variance share one parameter) such that all channels in $\mathbf{z}$ can be treated independently. Also the structure of Eq. 9 is well suited for a DNN implementation. In our experiments, we focus on this Bernoulli variant of the general result in Eq. (7). In the Supplement, we study a number of other relevant members of the EF. Furthermore, it is interesting to study the structure of Eq. (8) more carefully. The first term minimizes the mismatch between the forward and the posterior distribution with respect to the forward blocks. The second term is the uncertainty-weighted backward activation $b_{kj}$ which modulates local gradients (see Supplement). Therefore, the backward activations $b_{kj}$ act directly as learning signals for local updates. The BLL method is therefore related to feedback alignment [Lillicrap et al., 2020] and target propagation [Frenkel et al., 2021] where backward information is provided through random weights. However, since the gradients of the backward blocks appear in the second term, our model also provides a principled way to optimize the backward flow of information from the targets.

**Data mixing schedule:** The equation for the posterior distribution Eq. 9 contains a data mixing parameter $m$, with $0\leq m\leq 1$, that scales the influence of the backward messages in the posterior distribution. This parameter serves two important functions, (1) It scales the balance between forward and backward messages in the posterior distribution $\rho$ and (2) it scales the first term in the parameter updates Eq. 8. We found that a annealing schedule for this parameter that decreases $m$ slowly during learning works well in practice. If not stated otherwise, we used $m=(1+\tau\,M)^{-1}$ in our experiments, where $M$ is the index of the current epoch and $\tau$ is a scaling parameter (see the Supplement for further details).

## 4    Experimental results

We evaluated the BLL model on a number of vision and sequence learning tasks. All models used the Bernoulli BLL gradients described in Eq. (8) for local optimization. Additional details of the network models can be found in the Supplement.

### 4.1    Block-local learning of vision benchmark tasks

We compare the performance of our block local learning (BLL) algorithm with that of end-to-end backprop (BP) and Feedback Alignment (FA) Lillicrap et al. [2014b]. Three datasets are considered: MNIST, Fashion MNIST and CIFAR10 together with two residual network architectures [He et al., 2016]: ResNet-18 and ResNet-50, each trained with one of the three methods (BP, FA, BLL).

The BLL architectures were split into 4 blocks that were trained locally using the Bernoulli loss in Eq. (8). Splits were introduced after residual layers of the ResNet architecture by grouping subsequent layers into blocks. Group sizes were (4,5,4,5) for ResNet-18 and (12,13,12,13) for ResNet-50. Backward twin networks were here constructed simply by using the same network architecture (ResNet-18 or ResNet-50) in reverse order, introducing appropriate splits to provide intermediate targets. For CIFAR-10 gradients were propagated between two neighboring blocks (see Supplement for details and a comparison with purely local gradients). The kernels of ResNet-18/ResNet-50 + FA architectures used during backpropagation are fixed and uniformly initialised following the Kaiming He et al. [2015] initialisation method. The bias is set to one.

The results are summarized in Table 1. Test top-1, top-3 and train top-1 accuracies are shown. Top-3 accuracies count the number of test samples for which the correct class was among the network's

|  | **MNIST** | | | **Fahion-MNIST** | | | **CIFAR-10** | | |
|---|---|---|---|---|---|---|---|---|---|
|  | test-1 | test-3 | train-1 | test-1 | test-3 | train-1 | test-1 | test-3 | train-1 |
| ResNet-18 + BP | 99.5 | 100 | 99.7 | 92.7 | 99.3 | 96.0 | 95.2 | 99.3 | 100 |
| ResNet-50 + BP | 99.5 | 99.9 | 100 | 89.0 | 98.9 | 92.7 | 94.0 | 99.2 | 99.8 |
| ResNet-18 + FA | 99.0 | 99.9 | 100 | 87.9 | 98.6 | 92.1 | 70.4 | 92.5 | 80.9 |
| ResNet-50 + FA | 98.9 | 99.9 | 100 | 83.1 | 97.9 | 83.7 | 70.3 | 92.0 | 79.3 |
| ResNet-18 + BLL | 99.4 | 100 | 99.6 | 91.2 | 98.8 | 91.0 | 72.2 | 93.0 | 98.8 |
| ResNet-50 + BLL | 99.4 | 99.8 | 99.2 | 88.7 | 99.0 | 85.9 | 73.4 | 92.7 | 99.7 |

Table 1: Classification accuracy (% correct) on vision tasks. BP: end-to-end backprop, FA: feedback alignment, BLL: block local learning. Test-1, test-3 and train-1 represent the top-1, top-3 test accuracy and top-1 training accuracy respectively.

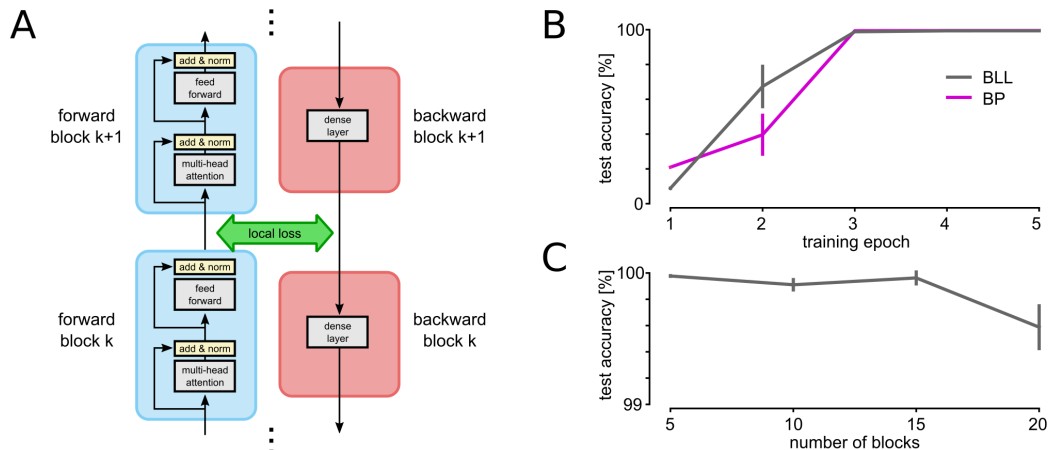

Figure 3: Block local learning of transformer architecture. **A:** Illustration of the transformer twin network. **B:** Learning curves of block local (BLL) and backprop (BP) training. **C:** Test accuracy vs. number of blocks in the transformer model. Error bars show standard deviations over 5 independent runs.

3 highest output activations. See Supplement for results over multiple runs. BLL achieved good performance on MNIST and Fashion-MNIST, closely matching end-to-end training and outperforming FA networks. Note that in contrast to FA and BP, BLL does not need to compute error gradients at the output but can work directly with the target labels. Performance on CIFAR-10 was significantly lower than BP but outperformed FA. Interestingly the performance on the training set was close to perfect for ResNet-50 suggesting over-fitting the task.

### 4.2 Block-local transformer architecture for sequence-to-sequence learning

Transformer architectures are in principle well suited for distributed computing due to their modular network structure that comprises a repetition of homogeneous blocks. We demonstrate a proof-of-concept result on training a transformer with BLL. We used a transformer model with 20 self-attention blocks with a single attention head each. Block local losses were added after each layer and blocks were trained locally. A backward twin network was constructed by projecting targets through dense layers and used the Bernoulli loss Eq. (8) for local training (see Fig. 3 A for an illustration). The transformer was trained on a sequence-to-sequence task, where a random permutation of numbers 0..9 was presented on the input and had to be re-generated at the output in reverse order. We trained the network for 5 epochs.

BLL achieves convergence speed that is comparable to that of end-to-end BP on this task. Fig. 3 B shows learning curves of BLL and BP. Both algorithms converge after around 3 epochs to nearly perfect performance. BLL also achieved good performance for a wide range of network depths. Fig. 3 C shows the performance after 5 epochs for different transformer architectures. Using only 5 transformer blocks yields performance of around 99.9% (average over five independent runs). The

test accuracy on this task for the 20 block transformer was 99.6%. These results suggest that the BLL method is equally applicable to transformer architectures.

## 5 Discussion

In this work, we have demonstrated a general purpose probabilistic framework for rigorously defining block-local losses for deep architectures. This not only provides a novel way of performing distributed training of large models but also hints at new paradigms of self-supervised training that are biologically plausible. We have also shown that our block-local training approach outperforms existing local training approaches while still getting around the locking and weight transport problems. Our method introduces a twin network that propagates information backwards from the targets to the input and automatically estimates uncertainties on intermediate layers. This is achieved by representing probability distributions in the network activations. The forward network and its backward twin can work in parallel and with different sets of weights.

The proposed method may also help further blur the boundary between deep learning and probabilistic models. A number of previous models have shown that DNNs are capable of representing probability distribution [Abdar et al., 2021, Pawlowski et al., 2017, Tran et al., 2019, Malinin and Gales, 2019]. Unlike these previous methods, our method does not require Monte Carlo sampling or contrastive training, but instead exploits the log-linear structure of exponential family distributions to efficiently propagate uncertainty-aware messages through a network using a belief-propagation strategy. We have demonstrated that implicit uncertainty messages can be learnt from sparse data and accurately represent the network's performance.

Greedy block-local learning has recently shown compelling performance on a number of tasks [Nøkland and Eidnes, 2019, Siddiqui et al., 2023]. These methods use local losses with an information-theoretic motivation but are agnostic to global back-propagating information. In future work, it may be interesting to combine these approaches with the proposed model to get the best of both worlds. Being able to produce block-level uncertainty predictions can also be useful for enhancing the sparsity of the network and using optimal amount of compute for predictions. The uncertainty predictions can also be used to handle missing labels, and for evaluating the model's confidence about its predictions. Since the framework is flexible enough to apply to self-supervised training, it can be used on unlabelled and multi-modal datasets as well. Due to the local nature of the training process, our method is particularly attractive for application on neuromorphic systems that co-locate memory and compute and use orders of magnitude less energy if the computation is local.

This work addresses potential problems of modern ML: The estimation of uncertainties in neural networks is an important open problem and understanding the underlying mechanisms better will likely help to make ML models safer and more reliable. Also the main focus of this work, which is on distributing large ML models over many compute nodes may make these model more energy efficient in the future. The energy consumption and resulting carbon footprint of ML is a major concern and the proposed model may provide a new direction to approach this problem. This method may enable training of larger models which also come with associated risks in terms of biases and inappropriate use in the real world. It is also not known what biases using this method itself and extensions with sparsity may introduce in the models predictions.

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
