# Supplementary Information: Block-local learning with probabilistic latent representations

## 1 A probabilistic formulation of distributed learning

### 1.1 Markov chain model

Here we provide additional details to the learning model presented in Section 3 of the main text. To establish these results we consider the Markov chain model $\mathbf{x} \to \mathbf{z}_1 \to \mathbf{z}_2 \to \cdots \to \mathbf{y}$ of a DNN with inputs $\mathbf{x}$, outputs $\mathbf{y}$ and intermediate representations $\mathbf{z}_k$ at block $k$. To simplify the notation we will define the input $\mathbf{z}_0 := \mathbf{x}$ and output $\mathbf{z}_N := \mathbf{y}$ layers, and $\mathbf{z} = \{\mathbf{z}_k\}, 1 \le k < N$, the auxiliary latent variables. A DNN $\mathcal{N}_A$ suggests a conditional independence structure given by the fully factorized Markov chain of random variables $\mathbf{z}_k$

$$p\left(\mathbf{y}, \mathbf{z} \mid \mathbf{x}\right) \;=\; p\left(\mathbf{z}_1 \ldots \mathbf{z}_N \mid \mathbf{z}_0\right) \;=\; \prod_{k=1}^{N} p_k\left(\mathbf{z}_k \mid \mathbf{z}_{k-1}\right) \;. \tag{S1}$$

The computation of messages $\alpha_k$ comes naturally in a feed-forward neural network as the flow of information follows the canonical form, input $\to$ output. Every block of the network thus translates $\alpha_{k-1} \to \alpha_k$ by outputting the statistical parameters of the conditional distribution $p\left(\mathbf{z}_k \mid \mathbf{x}\right)$ and takes $p\left(\mathbf{z}_{k-1} \mid \mathbf{x}\right)$ as input. This interpretation is viable for a suitable split of any DNN into $N$ blocks, that fulfils a mild set of conditions (see Section 1.3 for details). It is important to note that the random variables $(\mathbf{z}_1, \mathbf{z}_2, \dots)$ are only implicit. The network generates the parameters to the probability distribution and at no points needs to sample values for these random variables.

### 1.2 Using latent representations to construct probabilistic block-local losses

Many commonly used loss functions in deep learning have a probabilistic interpretation, e.g. the cross entropy loss of a binary classifier is identical to the Bernoulli log likelihood, and the mean squared error is up to a constant equivalent to the log-likelihood of a Gaussian with constant variance. In this formulation, the outputs of the DNN are interpreted as the statistical parameters to a conditional probability distribution (e.g. the mean of a Gaussian) and the loss function measures the support of observed data samples $\mathbf{x}$ and $\mathbf{y}$.

To introduce intermediate block-local representations $\mathbf{z}_k$ in the network we consider an upper bound to the log-likelihood loss (Eq. 1 of the main text)

$$\mathcal{L}_1 \;=\; -\log p\left(\mathbf{y} \mid \mathbf{x}\right) + \frac{1}{N} \sum_{k=1}^{N} \mathcal{D}_{KL}\left(q_k \mid p_k\right) \;, \tag{S2}$$

where $p_k$ and $q_k$ are true and variational posterior distributions over latent variables $p\left(\mathbf{z}_k \mid \mathbf{x}, \mathbf{y}\right)$ and $q\left(\mathbf{z}_k \mid \mathbf{x}, \mathbf{y}\right)$, respectively. Using the Markov property (S1) assuming a fully factorized distribution, implies the conditional independence

$$p\left(\mathbf{y}, \mathbf{z}_k \mid \mathbf{x}\right) = p\left(\mathbf{y} \mid \mathbf{z}_k\right) p\left(\mathbf{z}_k \mid \mathbf{x}\right) \;. \tag{S3}$$

28 Using this Eq. S2 becomes

$$
\begin{aligned}
\mathcal{L}_1 &= -\log p\left(\mathbf{y} \mid \mathbf{x}\right) + \frac{1}{N} \sum_{k=1}^{N} \mathcal{D}_{KL}\left(q_k \mid p_k\right) \\
&= \frac{1}{N} \sum_{k=1}^{N} \left\langle \log \frac{q\left(\mathbf{z}_k \mid \mathbf{x}, \mathbf{y}\right)}{p\left(\mathbf{y}, \mathbf{z}_k \mid \mathbf{x}\right)} \right\rangle_{q_k} \\
&= \frac{1}{N} \sum_{k=1}^{N} \left\langle \log \frac{q\left(\mathbf{z}_k \mid \mathbf{x}, \mathbf{y}\right)}{p\left(\mathbf{z}_k \mid \mathbf{x}\right)} - \log p\left(\mathbf{y} \mid \mathbf{z}_k\right) \right\rangle_{q_k} \\
&= \frac{1}{N} \sum_{k=1}^{N} \mathcal{D}_{KL}\left(\rho_k(\mathbf{x}, \mathbf{y}) \mid \alpha_k(\mathbf{x})\right) - \left\langle \log p\left(\mathbf{y} \mid \mathbf{z}_k\right) \right\rangle_{q_k} \, . \quad \text{(S4)}
\end{aligned}
$$

29 Eq. S4 is an upper bound on log-likelihood loss $\mathcal{L}^* = -\log p\left(\mathbf{y} \mid \mathbf{x}\right) \leq \mathcal{L}_1$. Since $\mathcal{L}^*$ is strictly
30 positive, minimizing $\mathcal{L}_1$ to zeros implies that also $\mathcal{L}^*$ becomes zero Mnih and Gregor [2014].

### 1.3 General exponential family distribution

32 To arrive at a result for the gradient of the first (KL-divergence) term in Eq. S4 we seek distributions
33 for which the marginals can be computed in closed form. We assume forward messages $\alpha$ and
34 posterior $\rho$ be given by general exponential family distributions

$$
\alpha_k\left(\mathbf{z}_k\right) = \prod_j \alpha_{kj}\left(z_{kj}\right) = \prod_j h(z_{kj}) \exp\left(T\left(z_{kj}\right)\phi_{kj} - A\left(\phi_{kj}\right)\right) \quad \text{(S5)}
$$

$$
\rho_k\left(\mathbf{z}_k\right) = \prod_j \rho_{kj}\left(z_{kj}\right) = \prod_j h(z_{kj}) \exp\left(T\left(z_{kj}\right)\gamma_{kj} - A\left(\gamma_{kj}\right)\right) \quad \text{(S6)}
$$

35 with base measure $h$, sufficient statistics $T$, log-partition function $A$, and natural parameters $\phi_{kj}$ and
36 $\gamma_{kj}$. Using this the KL loss becomes

$$
\mathcal{L}_V^{(k)} = \mathcal{D}_{KL}\left(\rho_k \mid \alpha_k\right) = \sum_j \left\langle T\left(z_{kj}\right)\left(\phi_{kj} - \gamma_{kj}\right) - A\left(\phi_{kj}\right) + A\left(\gamma_{kj}\right) \right\rangle_{\rho_{kj}}, \quad \text{(S7)}
$$

37 and thus

$$
\begin{aligned}
-\frac{\partial}{\partial\theta}\mathcal{L}_V^{(k)} = \sum_j & \left( \left\langle T\left(z_{kj}\right) \right\rangle_{\rho_{kj}} - \left\langle T\left(z_{kj}\right) \right\rangle_{\alpha_{kj}} \right) \frac{\partial}{\partial\theta}\phi_{kj} + \\
& \underbrace{\left( \left\langle T\left(z_{kj}\right)^2 \right\rangle_{\rho_{kj}} - \left\langle T\left(z_{kj}\right) \right\rangle_{\rho_{kj}}^2 \right)}_{\sigma^2(\rho_{kj})} \left(\phi_{kj} - \gamma_{kj}\right)\frac{\partial}{\partial\theta}\gamma_{kj}, \quad \text{(S8)}
\end{aligned}
$$

38 which by defining $\mu\left(p\right) = \left\langle T\left(z_{kj}\right) \right\rangle_p$ can be written in the compact form

$$
-\frac{\partial}{\partial\theta}\mathcal{L}_V^{(k)} = \sum_j \left(\mu\left(\rho_{kj}\right) - \mu\left(\alpha_{kj}\right)\right)\frac{\partial}{\partial\theta}\phi_{kj} + \sigma^2\left(\rho_{kj}\right)\left(\phi_{kj} - \gamma_{kj}\right)\frac{\partial}{\partial\theta}\gamma_{kj} \, .
$$

39 This is the result Eq. (7) of the main text.

#### 1.3.1 Example: Bernoulli random variables

41 For the example of a Bernoulli random variable we have $T\left(z_{kj}\right) = z_{kj}$, $A\left(\gamma\right) = \log\left(1 + e^\gamma\right)$,
42 $\left\langle T\left(z_{kj}\right) \right\rangle_{\rho_{kj}} = \rho_{kj}$, and furthermore $\sigma^2\left(\rho_{kj}\right) = \rho_{kj}\left(1 - \rho_{kj}\right)$. We get

$$
-\frac{\partial}{\partial\theta}\mathcal{L}_V^{(k)} = \sum_{k,j} \left(\rho_{kj} - \alpha_{kj}\right)\frac{\partial}{\partial\theta}\phi_{kj} + \rho_{kj}\left(1 - \rho_{kj}\right)\left(\phi_{kj} - \gamma_{kj}\right)\frac{\partial}{\partial\theta}\gamma_{kj} \, . \quad \text{(S9)}
$$

Using the ansatz $\phi_{kj} = a_{kj}$ and $\gamma_{kj} = a_{kj} + b_{kj}$, $\rho_{kj} = S(a_{kj} + b_{kj}) = p(z_{kj} = 1 \mid \mathbf{x}, \mathbf{y})$ with $a_{kj} = f_j(\mathbf{a}_{k-1})$ and $b_{kj} = g_j(\mathbf{b}_{k+1})$ we further get

$$-\frac{\partial}{\partial\theta}\mathcal{L}_V^{(k)} = \sum_{k,j} (\rho_{kj} - \alpha_{kj}) \frac{\partial}{\partial\theta} a_{kj} - \rho_{kj}(1 - \rho_{kj}) b_{kj} \left( \frac{\partial}{\partial\theta} a_{kj} + \frac{\partial}{\partial\theta} b_{kj} \right). \tag{S10}$$

For the Bernoulli case it is also easy to verify that our approach is sound. Here, the natural parameters are given by the logg-odds $a_{kj} = \log \frac{p(z_{kj}=1\mid\mathbf{x})}{p(z_{kj}=0\mid\mathbf{x})}$ and $b_{kj} = \log \frac{p(\mathbf{y}\mid z_{kj}=1)}{p(\mathbf{y}\mid z_{kj}=0)}$. Plugging this into the expression for $\rho_{kj}$ we get $\rho_{kj} = S(a_{kj} + b_{kj}) = S\left( \log \frac{p(z_{kj}=1\mid\mathbf{x})}{p(z_{kj}=0\mid\mathbf{x})} + \log \frac{p(\mathbf{y}\mid z_{kj}=1)}{p(\mathbf{y}\mid z_{kj}=0)} \right) = p(z_{kj} = 1 \mid \mathbf{x}, \mathbf{y})$.

### 1.3.2 Example: Gaussian random variables with constant variance

For the example of a Gaussian random variable with constant variance we have $T(z_{kj}) = z_{kj}$, $\left\langle T(z_{kj}) \right\rangle_{\rho_{kj}} = \phi_{kj}$, and furthermore $\sigma2(\rho_{kj}) = \sigma^2 (= const)$. We get

$$-\frac{\partial}{\partial\theta}\mathcal{L}_V^{(k)} = \sum_{k,j} (\gamma_{kj} - \phi_{kj}) \frac{\partial}{\partial\theta}\phi_{kj} + \sigma(\phi_{kj} - \gamma_{kj}) \frac{\partial}{\partial\theta}\gamma_{kj} \tag{S11}$$

Using the ansatz $\phi_{kj} = a_{kj}$ and $\gamma_{kj} = a_{kj} + b_{kj}$, we further get

$$-\frac{\partial}{\partial\theta}\mathcal{L}_V^{(k)} = \sum_{k,j} (1 - \sigma) b_{kj} \frac{\partial}{\partial\theta} a_{kj} - \sigma b_{kj} \frac{\partial}{\partial\theta} b_{kj}. \tag{S12}$$

### 1.3.3 Example: Poisson random variables

For the example of a Poisson random variable we have $T(z_{kj}) = z_{kj}$, $A(\gamma) = e^\gamma$, $\left\langle T(z_{kj}) \right\rangle_{\rho_{kj}} = e^{\gamma_{kj}}$, furthermore $\sigma^2(\rho_{kj}) = \rho_{kj} = e^{\gamma_{kj}}$ and $\alpha_{kj} = e^{\phi_{kj}}$. Using again $\phi_{kj} = a_{kj}$ and $\gamma_{kj} = a_{kj} + b_{kj}$, we get

$$-\frac{\partial}{\partial\theta}\mathcal{L}_V^{(k)} = \sum_{k,j} (\rho_{kj} - \alpha_{kj}) \frac{\partial}{\partial\theta} a_{kj} - \rho_{kj} b_{kj} \left( \frac{\partial}{\partial\theta} a_{kj} + \frac{\partial}{\partial\theta} b_{kj} \right). \tag{S13}$$

### 1.3.4 Estimating the log-likelihood loss through posterior mixing

Finally we show how the remaining term $\left\langle \log p(\mathbf{y} \mid \mathbf{z}_k) \right\rangle_{q_k}$ in Eq. S4 can be estimated locally. First we note that the $-\log p(\mathbf{y} \mid \mathbf{z}_k)$ is of the same form as the log-likelihood loss (Eq. (1) of the main text), i.e. the likelihood of the data labels $\mathbf{y}$ of the residual network $\mathbf{z}_k \rightarrow \mathbf{y}$. Thus treating $\mathbf{z}_k$ as block-local input data and minimizing the augmented ELBO loss from layer $\mathbf{z}_k \rightarrow \mathbf{z}_N$ minimizes another lower bound on the global loss $\mathcal{L}^*$. By inserting Eq. S4 recursively into itself we get

$$\mathcal{L}_2 = \frac{1}{N} \sum_{k=1}^{N} \left( \mathcal{D}_{KL}\left( \rho_k(\mathbf{x}, \mathbf{y}) \mid \alpha_k(\mathbf{x}) \right) + \right.$$
$$\left. \frac{1}{N-k} \sum_{l=k+1}^{N} \left( \left\langle \mathcal{D}_{KL}\left( \rho_l(\mathbf{z}_k, \mathbf{y}) \mid \alpha_l(\mathbf{z}_k) \right) \right\rangle_{q_k} - \left\langle \log p(\mathbf{y} \mid \mathbf{z}_l) \right\rangle_{q_k \rightarrow q_l} \right) \right), \tag{S14}$$

where we used the short-hand notation $\left\langle f(\mathbf{z}_l) \right\rangle_{q_k \rightarrow q_l} = \left\langle \left\langle f(\mathbf{z}_l) \right\rangle_{q_l} \right\rangle_{q_k}$. Note that the forward network is able to compute this expression since each block computes the required marginal locally by Eq. (3). That is, the data is augmented by choosing a block $k$ and instead of propagating $\alpha_k$ into block $k+1$ the posterior $\rho_k$ is propagated forward. By iterating another recursion we get

$$\mathcal{L}_3 = \frac{1}{N} \sum_{k=1}^{N} \left( \mathcal{D}_{KL}\left( \rho_k(\mathbf{x}, \mathbf{y}) \mid \alpha_k(\mathbf{x}) \right) + \frac{1}{N-k} \sum_{l=k+1}^{N} \left( \left\langle \mathcal{D}_{KL}\left( \rho_l(\mathbf{z}_k, \mathbf{y}) \mid \alpha_l(\mathbf{z}_k) \right) \right\rangle_{q_k} + \right. \right.$$
$$\left. \left. = \frac{1}{N-l} \sum_{l'=l+1}^{N} \left( \left\langle \mathcal{D}_{KL}\left( \rho_l(\mathbf{z}_k, \mathbf{y}) \mid \alpha_l(\mathbf{z}_k) \right) \right\rangle_{q_k \rightarrow q_l} - \left\langle \log p(\mathbf{y} \mid \mathbf{z}_{l'}) \right\rangle_{q_k \rightarrow q_l \rightarrow q_{l'}} \right) \right) \right).$$

67 This result implies a hierarchy of loss functions $0 \leq \mathcal{L}^* \leq \mathcal{L}_1 \leq \mathcal{L}_2 \leq ...$, where $\mathcal{L}_N$ consists only
68 of $\mathcal{D}_{KL}$-terms between forward messages $\alpha$ and posteriors $\rho$ that were generated by propagating
69 different paths $q_k \to q_l \to q_{l'} \to \ldots$ through the network. While this posterior mixing would be
70 computable in principle in our model, it turns out to be quite expensive since exponentially many
71 (exponential in the number of blocks $N$) such paths have to be considered.

72 We therefore used a different approach by introducing the mixing parameter $m$ in Eq. 8 to redefine
73 the posterior $\rho_{kj} = S\left(a_{kj} + m\, b_{kj}\right)$, and replacing in Eq. S10. Note that in the limit $m \to 0$ we
74 have $\rho_{kj} = \alpha_{kj}$ and therefore the posterior mixing described above can be omitted. We therefore
75 used small values $m$ and only include it in the loss as described in Eq. 8 of the main text. We found
76 that combining a suitable schedule that slowly anneals the mixing parameter $m$ towards zero during
77 training gives good results in practice. We used $m = \left(1 + \tau\, M\right)^{-1}$ in our experiments, where $M$
78 is the index of the current epoch and $\tau$ is a scaling parameter that was set to $\tau = 0.5$ if not stated
79 otherwise. In the transformer example in Fig. 3 we used a constant mixing $m = 0.01$ throughout
80 training.

# 2 Experimental procedure

## 2.1 Forward-backward networks as autoencoder

83 For the convolutional autoencoder in Section 3.3 of the main text we used a convolutional neural net-
84 work with 2 layers with leaky ReLu activation function for decoder and encoder. Batch normalization
85 was used after the convolution/deconvolution layers. Encoder network in addition used max-pooling
86 after each convolution layer. The bottleneck layer ($\mathbf{y}$) had 128 channels. Fashion MNIST images were
87 augmented with 28x28 pixel images as targets for the uncertainty outputs, giving a total input/target
88 size of 56x28. Uncertainty inputs/targets were set to a constant of 0.2 during training for all channels
89 and training samples.

90 Network output images were also split into 2 28x28 patches corresponding to training mean and
91 uncertainty channels. Let $\mu_n^*$ and $s_n^*$ denote mean and uncertainty channels of training sample $n$,
92 respectively, and let $\mu_n$ and $s_n$ be the corresponding network outputs. For training and testing we
93 used the Gaussian Kullback-Leibler divergence loss

$$\mathcal{L}_{\text{KL}} \;=\; \frac{1}{2\,M} \sum_{n=1}^{M} \left( s_n - s_n^* \;+\; \frac{e^{s_n^*} + \left(\mu_n^* - \mu_n\right)^2}{e^{s_n}} - 1 \right) \;, \tag{S15}$$

94 where $M$ is here the number of training samples and $s_n$ corresponding to log variances. The Adam
95 optimizer with learning rate of 0.001 was used for training. For validation to further assess the
96 mismatch between estimated and true prediction errors in Fig. 2 of the main text, we also used the
97 MSE matching loss

$$\mathcal{L}_{\text{MM}} \;=\; \frac{1}{M} \sum_{n=1}^{M} \left( \left(\mu_n^* - \mu_n\right)^2 - e^{s_n} \right)^2 \;, \tag{S16}$$

98 that estimates the distance between the empirical MSE of predictions, and the MSE estimator loss

$$\mathcal{L}_{\text{ME}} \;=\; \frac{1}{M} \sum_{n=1}^{M} s_n \;, \tag{S17}$$

99 that is a global uncertainty estimator (mean variance predicted by the network). Uncertainty outputs
100 in Fig. 2B were clipped to min and maximum range for the 5 examples given and presented as
101 grayscale images.

## 2.2 Block-local learning with vision benchmark tasks

103 BLL Architectures used in Section 4 were adapted from ResNet-18 and ResNet-50 architectures.
104 Batch normalization was used after the convolution layers as is standard for ResNet architectures.
105 These networks were split into 4 blocks that were trained locally. Backward twin networks were
106 constructed using the same network in reverse order, again split into 4 blocks to provide intermediate
107 losses. The ResNet-18, for example, with its group sizes (4,5,4,5) was reversed into a group sizes
108 of (5,4,5,4). Any convolution in the forward network with a stride more that 1 (i.e, Downsampling)
109 was appended with an Upsampling layer of same stride in the backward network. Gradients were
110 blocked after every layer in forward and backward networks and auxiliary losses (Eq. (8) of the main
111 text) added for block local learning. For CIFAR10 experiments, additional tests were conducted with
112 stopping gradients only after every two neighboring blocks.

| | **MNIST** | | |
| --- | --- | --- | --- |
| | test-1 (mean±std) | test-3 (mean±std) | train-1 (mean±std) |
| ResNet-18 + BP | 99.5±0.1 | 99.9±0.01 | 99.9±0.03 |
| ResNet-50 + BP | 99.5±0.06 | 99.9±0.0 | 99.9±0.1 |
| ResNet-18 + FA | 98.5±0.1 | 99.9±0.03 | 99.6±0.1 |
| ResNet-50 + FA | 98.9±0.06 | 99.9±0.03 | 100±0.0 |
| ResNet-18 + BLL | 99.3±0.1 | 100±0.0 | 99.5±0.3 |
| ResNet-50 + BLL | 99.1±0.4 | 99.9±0.1 | 99.2±0.2 |

Table 1: Classification accuracy (% correct) for 5 runs on MNIST vision tasks. BP: end-to-end backprop, FA: feedback alignment, BLL: block local learning. Test-1, test-3 and train-1 represent the top-1, top-3 test accuracy and top-1 training accuracy respectively.

| | **Fahion-MNIST** | | |
| --- | --- | --- | --- |
| | test-1 (mean±std) | test-3 (mean±std) | train-1 (mean±std) |
| ResNet-18 + BP | 92.7±0.1 | 99.2±0.7 | 99.3±0.1 |
| ResNet-50 + BP | 92.3±0.3 | 99.3±0.1 | 99.0±0.1 |
| ResNet-18 + FA | 88.2±0.3 | 98.7±0.2 | 94.3±0.8 |
| ResNet-50 + FA | 86.6±0.7 | 98.6±0.1 | 91.1±2.2 |
| ResNet-18 + BLL | 90.0±1.2 | 99.0±0.2 | 90.7±2.9 |
| ResNet-50 + BLL | 86.9±1.3 | 98.4±0.4 | 85.9±1.1 |

Table 2: As in Table 1. Classification accuracy (% correct) for 5 runs on FashionMNIST vision tasks.

### 2.2.1 MNIST and FashionMNIST vision tasks

MNIST images were pre-processed by normalization to mean 0 and stds 1. FashionMNIST images were in addition augmented with random horizontal flips. MNIST is a freely available dataset consisting of 60,000 + 10,000 (train + test) grayscale images of handwritten digits published under the GNU General Public License v3.0. FashionMNIST is a freely available dataset consisting of 60,000 + 10,000 (train + test) grayscale images of fashion items published under the MIT License (MIT) [Xiao et al., 2017]. After the submission of the main paper we ran additional trials with FA that gave better results on Fashion-MNIST and CIFAR10, which were included in Table 2 and will be added in the main paper after the revision. Overall we found the trial-by-trial variability of FA high compared to other methods analyzed.

### 2.2.2 CIFAR10 vision task

The BLL networks for CIFAR10 experiments also used the ResNet architectures as described in Section 2.2. However the gradients were propagated in between two neighbouring blocks instead of single block. This resulted in slightly better performance in our experiments, see Table 3. We used SGD optimizer with a learning rate of 0.002 and a momentum of 0.9. Additionally, we used a Cosine annealing learning rate scheduler [Loshchilov and Hutter, 2017] with max iterations set to 140. The batch size was chosen to be 128 to maximize GPU utilization. We performed minimal hyperparameter ( Learning rate, LR scheduler $T_{max}$) tuning to obtain current results.

### 2.2.3 Feedback alignment

Resnet-18 and Resnet-50 architectures were also adapted for training with Feedback Alignment Lillicrap et al. [2014], for comparison. To do so, random and fixed kernels **B**, were used during backpropagation, while different ones, **W**, were used during the forward pass. Only **W** were updated and learned. Both kernels were of the same dimensionality (*output_channel*, *input_channel*, *Kernel_Width*, *Kernel_Height*) at each layer. Kernels were uniformly initialised using the Kaiming He et al. [2015] initialisation method. The bias term was set to one.

### 2.3 Hardware and software details

Most of our experiments were run on NVIDIA A100 GPUs and some initial evaluations and the MINST experiments were conducted on NVIDIA V100 and Quadro RTX 5000 GPUs. In total we used about 90,000 computational hours for training and hyper-parameter searches. ResNet18 and

| | CIFAR-10 | | |
|---|---|---|---|
| | test-1 | test-3 | train-1 |
| | (mean±std) | (mean±std) | (mean±std) |
| ResNet-18 + BP | 92.5±1.5 | 98.3±0.3 | 99.1±0.1 |
| ResNet-50 + BP | 91.1±1.1 | 98.7±0.2 | 98.1±0.9 |
| ResNet-18 + FA | 72.0±0.6 | 92.8±0.1 | 81.2±2.2 |
| ResNet-50 + FA | 62.5±0.4 | 88.2±0.2 | 66.9±1.1 |
| ResNet-18 + BLL (1) | 61.3±0.89 | 88.0±0.45 | 62.5±0.09 |
| ResNet-50 + BLL (1) | 59.9±1.02 | 87.8±0.27 | 62.6±1.07 |
| ResNet-18 + BLL (2) | 72.2±0.14 | 93.0±0.09 | 98.8±0.14 |
| ResNet-50 + BLL (2) | 73.4±0.47 | 92.7±0.28 | 99.7±0.06 |

Table 3: As in Table 1. Classification accuracy (% correct) for 5 runs on CIFAR10 task. BLL (x): block local learning with gradients propagated between x neighbouring blocks.

ResNet50 models and experiments were implemented in PyTorch [Paszke et al., 2019]. Transformer model for sequence-to-sequence learning was implemented in JAX [Bradbury et al., 2018].