# OpenReview forum: "Block-local learning with probabilistic latent representations"
_NeurIPS.cc/2023/Conference — Submitted to NeurIPS 2023_

### Official Review · Reviewer_xCFC · 2023-07-03

**Soundness:** 2 fair
**Presentation:** 1 poor
**Contribution:** 1 poor
**Rating:** 2
**Confidence:** 5

**Summary:**

The present work proposes a block-wise learning strategy, whereby the architecture is split into several blocks, with each block receiving an error signal stemming from a local (block-wise) loss. As this technique makes use of a parametrized twin network to compute these error signals, it also bypasses the so-called "weight transport problem" of the standard backprop algorithm. The motivation of this work is the hardware-friendliness of the resulting algorithm.

More precisely:

- Section 3.1 formulates the learning objective in a probabilistic fashion (likelihood maximization) and how to do it (variational inference)
- Section 3.2 explains the parametrization of the base model $p(z|x) := \alpha(z)$ to be trained and that of the variational posterior $q(z|x, y)$. The variational posterior itself decomposes into the product of  bottom-up top-down messages: $q(z|x, y) \propto p(z|x) p(y|z) := \alpha(z) \times \beta(z)$. The base model and the variational posterior are parametrized within the *exponential family* and the neural networks are made to output the natural parameters associated with these distributions. No stochastic quantity is ever propagated through the nets -- e.g. it does not operate as a VAE for instance.
- Section 3.3 grounds the previous idea in an encoder-decoder setting where the last layer of the decoder is augmented with extra channels to output pixel-wise variances in the image space. Training this architecture on F-MNIST with the proposed approach (the learning objective is not yet introduced at this stage), it is shown that the resulting uncertainties are good proxy to the reconstruction error.
- Section 3.4 introduces the proposed local (block-wise) learning objectives -- the derivation is in Appendices 1.2 -> 1.3.4. Heuristically: the learning rule reads as the forward/posterior mismatch backpropagated into the feedforward block (first term and second term of Eq. 7) and its feedback counterpart (second term of Eq. 7) through the natural parameters gradients. Importantly, an heuristic called "data mixing" is introduced to midly take into account the residual terms of the upper bound of the loss (details in Appendix).
- Section 4.1 presents results obtained on MNIST, F-MNIST and CIFAR-10 with the proposed technique on ResNet architectures, benchmarked against feedback alignment and standard backprop. The proposed approach performs comparably with backprop on MNIST and F-MNIST, slightly outperforms FA on CIFAR-10 but is considerably degraded with respect to backprop.
- Section 4.2 finally presents results on a 20-layers deep transformer architecture on a toy task (reverting a random permutation of numbers ranging between 0 and 9) where the proposed approach is shown to perform comparably with backprop.

**Strengths:**

- The paper tackles an interesting problem: block-wise local training casted into a probabilistic setting.
- The idea of the paper is interesting: the starting point is the same as Predictive Coding (Whittington & Bogacz, 2017), but: i/ picking a different variational family ii/ amortizing inference with a single forward pass (rather than minimizing energy for the E-step at each batch iteration) iii/ having a whole block of layers (where backprop applies) to compute the parameters of the distributions. I like the idea of stitching several algorithms together to solve a problem.

**Weaknesses:**

- The writing and the structure of the paper make it extremely difficult to deeply understand the proposed approach. To my eyes, ideas do not appear in the right order in the main, important ideas are in the appendix and the notations are confusing, important details are missing.
- It is *not* true that the algorithm parallelizes the forward and backward pass (L.40: "forward and backward propagation can work in parallel"). The underlying algorithm is just variational inference applied to a generative model conditioned on a given input $x$. I do not see any valid (theoretically grounded) argument as to why the first block could start processing a novel input $x'$ while the upper blocks process an input $x$. **The block-wise locality of the learning rule does not suffice as an argument here**.
- In the same vein, it is *neither* true that the algorithm allows for the parallelization the backward pass *across different blocks*. Here again, as the underlying algorithm simply is variational inference, I see no valid argument as to why each (feedforward) block could do without having a top-down error signal. However, it is true that the training of the *feedback* parameters can be parallelized.
- The derivation of the learning rule -- which should be, to my eyes, the *central piece* of the paper -- is unfortunately not presented in a sufficiently clear and detailed fashion.
- Section 3.3 on uncertainty estimation is weak and orthogonal to the scope of the paper in my eyes. The uncertainty estimation is carried out on a single simple task, and does not abide by the standard of the uncertainty estimation literature: are the uncertainties well calibrated? Can they be used for anomaly detection? What is the quantitative performance of the anomaly detection in terms of binary classification metrics? A good task to consider (and not too difficult) is the MVTech dataset (https://www.mvtec.com/company/research/datasets/mvtec-ad).
- The baselines chosen in the experimental section are not very relevant. The proposed technique applies to block-wise training, but not a single block-wise training baseline (e.g. Belilovsky (2019)) is considered. Why? A relevant choice would have been to consider a VGG-11 architecture with 3 layers per block and try to reach $\approx 67.6 \\%$ top-1 performance on ImageNet (Belilovsky et al, 2019: https://arxiv.org/pdf/1812.11446.pdf).
- Given that the use of backprop is allowed within each block and that ResNet architectures are considered, the experimental results are disappointing: the performance obtained on CIFAR-10 is very poor ($\approx 70 \\%$ accuracy on CIFAR-10 is achievable by a 4 layers-deep convnet), ImageNet32 (or even ImageNet) has not been considered, and some results are surprising (see one of the questions below).


**Questions:**

- It is more a suggestion for clarity improvement than a question: for the sake of clarifying the derivation of the learning rule, could you consider presenting things **in the main** closer to the following:
  + $-\log p(y|x) \leq L_b^k  (^*)$ with $L_b^k:=-\mathbb{E}_q[\log p(z^k, y|x)] - H(q(z^k |x,y)) = -\log p(y|x) + KL(q(z^k |x,y) || p(z^k|x,y))$.
  + Sum $(^*)$ over k, divide per $N$ to obtain the upper bound $L_b := -\log p(y|x)+ \frac{1}{N}\sum_{k=1}^N KL(q(z^k |x,y) || p(z^k|x,y))$.
  + Re-write $L_b$ as: $L_b=\frac{1}{N}\sum_{k=1}^N\ell(x, y, z^k)$ with $\ell(x, y, z^k) := \mathbb{E}_{q(z^k|x,y)}\left[\log \frac{q(z^k|x,y)}{p(y|z^k)p(y|z^k)}\right]$
  + Finally rewrite $\ell(x, y, z^k) = KL(q(z^k|x,y) || p(z^k|x)) - \mathbb{E}_{q(z^k|x,y)}\left[\log p(y|z^k)\right]$

For next questions onward, we denote $\ell^{(1)}(x, y, z^k) := KL(q(z^k|x,y) || p(z^k|x))$ and $\ell^{(2)}(x, y, z^k) := - \mathbb{E}_{q(z^k|x,y)}\left[\log p(y|z^k)\right]$ such that $\ell(x, y, z^k) = \ell^{(1)}(x, y, z^k) + \ell^{(2)}(x, y, z^k)$.

- In the light of the previous remark, I don't think it is optimal for clarity purposes to directly give the formula in Eq. (7). Also, Eq. 7 is essentially skewed as you simply write $\partial_\theta \ell^{(1)}(x, y, z^k)$ and *not* $\partial_\theta \ell (x, y, z^k)$ (i.e. the total ELBO).
Another thing which is very unclear is that you make no distinction between the parameter of the *base network* and that of the *target* / feedback network. In L. 104, you define $\theta$ as the "network parameters", but in Eq. 8, you are explicitly taking gradient of the output of the feedback network with respect to $\theta$. There is an ambiguity here that hinders clarity. Please distinguish between the two sets of parameters.

- Following up on the previous bullet: there is *no* explanation in the main of why you are discarding $\ell^{(2)}(x, y, z^k)$ from the ELBO gradient. It only when looking at 1.3.4 inside the supplementary material that we understand that: 1/$\ell^{(2)}(x, y, z^k)$ is intractable 2/ "data mixing",which appears to be a heuristic optimization trick in the main, is in fact intended to approximate/emulate the intractable contribution of $\ell^{(2)}(x, y, z^k)$. Even after reading multiple times L. 72-80, I still misunderstand this trick. If I missed something important, please clarify it inside the main.

- I'm doubtful about the experimental results: why does the proposed technique overfits so well CIFAR-10 while being so poor at generalization, in spite of using a ResNet? Do you train all layers (4 layers would be enough to overfit CIFAR-10)? Do you not apply optimization tricks to avoid overfitting (i.e. weight decay, dropout, data augmentation)? This is very surprising.

- One other question which is absent in the paper (perhaps hinted by Fig. 1) is how **the last block**, with the classification error signal, is trained? My assumption is that it is trained by mere backprop.

- **A potential interpretation for your CIFAR-10 results**. If the previous point holds true, my interpretation of your surprising CIFAR-10 results is that you might end up training *only* the last block (4 layers for ResNet-18), which is sufficient to overfit CIFAR-10, but generalizes as a 4 layers-deep architecture (consistently with my remark above) because the error signal received by previous layers might be irrelevant. To sanity-check this, I would suggest performing a low-dimension project (e.g. using t-SNE) of the activations of the last layer of the penultimate block (e.g. $a_2$ on Fig. 1) and visualize whether the classes are well separated. I assume (but perhaps I'm wrong) they are not.

- **Why the previous blocks might not learn?** This is something to be investigated further. I see several possibilities:
   + The "data mixing" trick is too heuristic and does not suffice to "emulate"/take into account the intractable part of your ELBO gradient.
   + If you indeed parallelize gradient computation *across different blocks*, it might be that some blocks never receive top-down error information.

- A really minor point at this stage would be to mention Predictive Coding (Whittington & Bogacz, 2017), whose starting point is the same as yours (e.g. variational inference on a generative model), but using a much simpler variational family.

- Coming back on uncertainty estimation: I'm not sure it is a desirable property that the model exhibit high pixel-wise variance for *in-distribution* features. A more desirable property rather is to have high uncertainty on *out-of-distribution* features -- namely: segmentation anomalies/defects. Although I still think this direction is orthogonal to the scope of the paper and should be removed, if this is of interest to you, consider the MVTech dataset and see if the pixel uncertainties can be leveraged to detect object anomalies.

- Another minor point: Frenkel et al (2021) does not suffice as a reference to Target Prop algorithms. Could you please add Lee et al (2015) -- the seminal target prop paper --, Meulemans et al (2020), Ernoult et al (2022).


**Limitations:**

To summarize my points of advice above:
- In terms of presentation, I would recommend:
   + you clarify the derivation of your learning rule along the lines suggested above,
   + state clearly the intractability of $\ell^{(2)}$ for the ELBO gradient and better explain the heuristic used ("data mixing"),
   + distinguish between the feedback and feedforward block parameters, e.g. $\theta_f$ and $\theta_b$,
   + remove that your approach allows for forward/backward pass parallelization and backward pass parallelization across layers. While you *can* do this in practice, the theoretical approach itself (e.g. variational inference) does not prescribe doing this. This might explain why the penultimate and upstream blocks don't learn.
+ **Please write a detailed pseudo-algorithm** for the proposed procedure for a given training batch. That would be extremely helpful.

- Try to check if the previous blocks are really learning. My hypothesis is that it is not and that it requires fixing the algorithm itself.

- If uncertainty estimation matters to you, I would suggest considering the MVTech dataset.

---

> ### Author Rebuttal · Authors · 2023-08-09
>
> We thank the reviewer for the vary detailed and valuable feedback. We made multiple changes to make the paper more accessible for a broad audience. We also clarified that the proposed method in fact combat the locking problem by adding pseudo code and additional explanations. We will provide additional details to individual concerns below.
>
> **Response to "weaknesses":**
>
> 1) The writing and the structure of the paper ...
>
> *Response:* Based also on comments by other reviewers we updated the theory part of the paper. We shortened some parts of the uncertainty estimation example in Fig.2 to make more space to describe the details of the model better. We also added additional details to the supplement (see Sections 3.1, 3.4, S1.3, S1.3.4 and S1.4).
>
> 2) It is *not* true that ...
>
> *Response*: The proposed BLL model is not just any application of variational inference (VI), but a very specific application of VI that allows us to separate the forward and backward propagation. The key point is that the linear combination of forward and backward messages in Eq.S10 form the variational posteriors, which reflects the conditional independence structure of (S3). This property allows us to split inference paths and parameter spaces, but is usually not exploited in other VI models. Hence, forward and backward messages can start propagating in parallel from both ends (our claim in L.40), and parameter updates can be computed as soon as both streams of information arrive at individual blocks. In contrast, in standard error back-propagation, backward updates are completely stale until losses are computed at the end of the forward pass. We added additional details to Sec. S2.2 to clarify and explain this feature of our algorithm in greater detail.
>
> 3) In the same vein, it is *neither* true that ...
>
> *Response:* We put more emphasis on the derivation as outlined above.
>
> 3) Section 3.3 on uncertainty estimation is weak ...
>
> *Response:* This example was meant for didactic reasons and we think that the it helps to better understand the model. We will consider the MVTech dataset to evaluate uncertainty detection in future work.
>
> 4) The baselines chosen in the experimental ...
>
> *Response:* Belilovsky et al. 2019, uses a a block-wise learning approach, but uses greedy learning where blocks are trained subsequently (rather than simultaneously). While this is somewhat orthogonal to our approach, where all parts of the network are trained in parallel, we agree that it is interesting to point out the existing different approaches. We included this and other baselines as suggested.
>
> 5) Given that the use of backprop is allowed ...
>
> *Response:* After the submission we found that the poor results in CIFAR-10 and Fashion MNIST were due to an error in the implementation. The error concerned the generation of splits between blocks and local losses and prevented meaningful learning in all but the last blocks. We re-ran all experiments and get  significantly better results now. We included these experiments in the updated version. We will also run additional experiments on ImageNet as suggested for the camera-ready version.
>
>
> **Response to "Questions":**
>
> 1) It is more a suggestion for clarity improvement ...
>
> *Response:* We restructured the math as suggested.
>
> 2) In the light of the previous ...
>
> *Response:* Our initial idea was to treat the model for the general case where parameter spaces are not separated. We see now that this only led to confusions and made this point clear from the beginning as suggested.
>
> 3) Following up on the previous ...
>
> *Response:* Yes, the data mixing as presented is a heuristic based on the recursive decomposition of the VI gradient. A theoretically exact version of data mixing is possible but produces exponentially many terms in the number of splits. For small number of splits this is still tractable but our first experiments suggested the much cheaper approximate version included in the paper suffices. We clarified this.
>
> 4) I'm doubtful about the experimental results ...
>
> *Response:* This difference is gone now after fixing the code as outlined above. We updated the results.
>
> 5) One other question which is absent in ...
>
> *Response:* Exactly. For the last block the exact gradients are straightforward.
>
> 6) A potential interpretation for your CIFAR-10 results ...
>
> *Response:* We added this analysis to visualize the parameter space in Fig. S1.
>
> 7) Why the previous blocks might not learn ...
>
> *Response:* The newly added analysis on t-SNE suggests that gradients do propagate through the network. We are investigating the impact of the data mixing heuristic, but a complete analysis seems to extend beyond the scope of the paper (or at least were not ready for this rebuttal). We are implementing the more complex recursive approximations L2, L3, … to see if they give significantly better results on tasks like CIFAR10, ImageNet, etc. We will include first results in the camera ready version of the paper.
>
> 8) A really minor point ...
>
> *Response:* We added a mention of and relation to predictive coding as suggested here and also by other reviewers as well.
>
> 9) Coming back on uncertainty estimation ...
>
> *Response:* Thank you, we will consider this data set.
>
> 10) Another minor point ...
>
> *Response:* We added these references.

---

> > ### Comment · Reviewer_xCFC · 2023-08-10
> > **Post-rebuttal answer**
> >
> > I hereby acknowledge I thoroughly read your rebuttal and took into account the updated version of the paper.
> >
> > **Responses to weaknesses**
> >
> > 1. OK
> >
> > 2.  *"Hence, forward and **backward messages** can start propagating in parallel from both ends (our claim in L.40), and **parameter updates can be computed as soon as both streams of information arrive at individual blocks***. So you acknowledge that you *cannot* update the **feedforward** parameters until a "backward" stream of information reaches "individual blocks": **therefore feedforward updates are locked**, thank you for acknowledging this point.  This being said, I do agree that **feedback** parameter updates are not locked. I already pointed this out in my initial review: *please refine the discussion about update locking separating out clearly the cases for feedforward and feedback parameters*.
> >
> > 3. Same.
> >
> > 3. Yet, it would still deserve further explorations and is still, to my eyes, orthogonal to your work. I would rather picture this part in Appendix.
> >
> > 4. Again: in spite of the peculiar topology of the underlying probabilistic model, your approach does *not* prescribe updating the *feedforward parameters* in parallel - or at least you haven't yet convinced me about this. So Belilovsky et al (2019), which embraces the sequentiality of the feedforward updates, stands as a relevant comparison to your work.
> >
> > 5. Thank you for acknowledging that my interpretation of your initial results was exact. Also, I really appreciate the amount of effort here. This being said, 84.2\% test-1 accuracy on CIFAR-10 and *most importantly* 87.6\% train-1 accuracy is still low and accounts for an **optimization issue**: you should be able to have $\approx 95\%$ train-1 accuracy. Therefore, there is still an unresolved issue here which I think might be due to *the bias induced by the estimation of the intractable term of your gradient formula*, or because the parallelization of your feedforward weight updates carries your feedforward weights in irrelevant directions.
> >
> > **Responses to questions**
> >
> > 1. Thank you, it is highly appreciated.
> >
> > 2. The update locking narrative depends on whether you consider feedforward or feedback parameters, so it is **crucial** to distinguish them.
> >
> > 3. OK, making this clearer is really important. Thank you.
> >
> > 4. I already addressed this point above.
> >
> > 5. OK, it is important to mention it explicitly somewhere.
> >
> > 6. *"The error concerned the generation of splits between blocks and local losses and prevented meaningful learning in all but the last blocks."*. Thank you for acknowledging my interpretation of your results.
> >
> > 7. This answer holds *after your code fix*. As to the results currently standing in the updated pdf: as I said above, I'm convinced that optimization is working well.
> >
> > 8. Thank you.
> >
> > 9. OK
> >
> > 10. OK
> >
> > Bottom line:
> > - Thank you a lot for the tremendous amount of effort put into your updated version. I really appreciate it.
> > - Still, I still strongly dispute the claim that you can parallelize feedforward parameter updates for the aforementioned reasons. I'm still deeply unconvinced about it.
> > - While the updated experimental results are better than the previous ones (where the bug is *exactly* the one I suspected), the train accuracy is still low, which hints at an **optimization issue** which, I hypothesize, might be due to: 1/ updating your feedforward weights in irrelevant directions as you update them in parallel *while you should await for a top-down error information signal*, as you *yourself* acknowledge, or 2/ the way you treat the intractable term of your gradient formula.

---

> > > ### Author Response · Authors · 2023-08-11
> > > **Reply to post rebuttal answer**
> > >
> > > 2.) The backward network only needs the (typically very sparse) labels to compute the required b_k terms and not the errors. Therefore, the activations of the backward network can be calculated in parallel to the forward network. Which means the weight updates can be calculated as soon as the forward pass in the forward network is done since the backward activations have already been calculated. Whereas in standard backprop, the backward pass can only begin once the forward pass is done i.e. it is locked. For a supervised training method, updating the parameters of each block with greater parallelism than this introduces a tradeoff in terms of staleness of parameters. This is because, information from the labels will always be required. Given these two constraints, we think we have achieved very high parallelism, so for the class of models that use tunable backward weights, there is not more that can be done.
> > >
> > > 5.) We added Belilovsky et al (2019) as a comparison despite it being a somewhat different approach as described in the rebuttal.
> > >
> > > 6.) We have very carefully checked the implementation now, we don’t think there are any more issues with it. Also our analysis demonstrates that gradients propagate throughout blocks suggesting that propagation of backward information works. Clearly the performance is below that of end-to-end training, but also previous approaches have reported lower training losses on CIFAR-10. As pointed out we are running ablation studies and augmentations to the model to further improve the performance (also on ImageNet). They look quite promising and will be included in the camera ready version.

---

> > > > ### Comment · Reviewer_xCFC · 2023-08-11
> > > > **About "parallelization"**
> > > >
> > > > 2) OK. I understand, mechanistically / algorithmically, what you mean by "parallelization" in your context of study: you can start the backward pass before the forward pass has ended. Yet **this comes at a cost**: this is because you *neglect* (or provide a heuristic treatment) for the intractable term of the ELBO, namely the gradient of $\mathbb{E}_{q_k}[\log p(y|z_k)]$, which entangles the feedforward and feedback pathways. In fact, this term essentially captures how much your approximate posterior accounts for the ground-truth labels, given the feedforward hidden activation $z_k$ - which therefore "locks" together the feedforward and feedback pathway. So replacing the exact, rigorous treatment of this contribution by the "data mixing" heuristic might in fact well unlock the backward pathway from the feedforward pathway, *algorithmically speaking*. Yet, from looking at your updated results on CIFAR-10, **this does not seem good enough a heuristic: 18\% test error on CIFAR-10 with a ResNet-18 is very high!** The bottom line is: you either need a very rigorous theoretical treatment (which is not the case here because of the intractable term of the ELBO), or a heuristic that really scales to complex tasks (which is neither the case here). This is an aside which goes beyond the scope of your paper, but a compelling example is BYOL (Grill et al, 2020): it is a non-contrastive self-supervised technique with no theoretical groundings but which was shown to scale comparably to SimCLR.
> > > >
> > > > 5) OK, I understand this in the light of your previous answer. Yet, you don't do as well as them.
> > > >
> > > > 6) *"previous approaches have reported lower training losses on CIFAR-10"*: which approaches? I can see them reported anywhere in your tables. Also, see my previous comment at 2) regarding your current results on CIFAR-10.
> > > >
> > > > *"They look quite promising and will be included in the camera ready version."*: the camera-ready version is only requested upon acceptance.

---

### Official Review · Reviewer_HQhi · 2023-07-06

**Soundness:** 2 fair
**Presentation:** 2 fair
**Contribution:** 3 good
**Rating:** 6
**Confidence:** 3

**Summary:**

The authors present a block-local learning rule as an alternative to end-to-end gradient backpropagation to train neural networks. They present a probabilistic view of neural network representations and assuming an exponential family of distributions, derive a learning rule that can be understood as forward and backward message passing between blocks. Notably their message passing interpretation allows them to formulate auxilliary local losses that can be then optimized using gradient descent at the block-local level. Furthermore, they claim that their algorithm is a more principled way of performing algorithms like Feedback Alignment and Target Propagation. Finally, they demonstrate that their algorithm can be used to train ResNets (ResNet-18 & ResNet-50) on certain vision datasets and Transformer architecture on a sequence prediction task. Overall, the proposed algorithm seems a promising alternative to backpropagation with local learning properties that enable better memory footprint and neuroscientific realism than backpropagation.

**Strengths:**

1. The paper does a good job in explaining the probabilstic interpretation and message passing view of forward and backward phase in neural networks.
2. The authors also empirically demonstrate the efficacy in training neural networks in Autoencoder setting as well as image classification and sequence prediction settings.
3. The proposed probabilistic interpretation of activations allows uncertainty estimation in neural network predictions. Although these uncertainty estimates are not benchmarked in the paper, I feel this is a strength of the proposed method.
4. Owing to its block-local nature, the proposed algorithm can be thought to be a biologically-plausible credit assignment algorithm for hierarchical neural networks. In doing so, this paper also offers a potential solution to memory-efficient distributed training of neural networks.

Overall, I think it's a strong submission and is very relevant for the NeurIPS community.

**Weaknesses:**

1. The writing and presentation in the paper is sometimes hard to read and understand, thereby leading to lack of an in-depth understanding of the proposed algorithm.
2. The current version of the paper does a great job in introducing the probabilistic interpretation of the neural network activations and the message passing view of forward and backward passes. However, their main learning rule is described in Section 3.4 which is not as clearly described. Unfortunately, the reader is deferred to the Appendix for key details of the derived learning rule, which adds to the lack of clarity regarding their proposed learning rule.
3. In contrast to the Feedback alignment algorithm, the block-local learning algorithm seems to overfit significantly to the Cifar-10 dataset. Although this is an interesting finding in itself, the paper doesn't offer an explanation into this phenomenon or which components of their algorithm contribute to this.
4. The discussion section offers more conjectures and falls short of discussing the implications of their results. For instance, the discussion section doesn't revisit the issue of overfitting or dive deeper into strategies around choosing the blocks & their backward counterparts and how these choices could affect the performance of the algorithm. The discussion section could also potentially highlight the potential biological plausibility of the proposed algorithm.

**Questions:**

1. In Like 115 (top of page 4), there seems to be a $log$ missing after the equals sign. Also, it seems that the right hand side of the expression would be a log-sum term. It is not clear how you write it as a sum of log terms in right hand side of Eq. 2.
2. In Eq. 4, you describe $\beta_k(z_k)$ to be the backward messages and $\rho_k(z_k)$ to be the estimated posterior. However, in Eq. 7, you describe passing posterior messages $\rho_k(z_k)$ using backward network activations. Could you please clarify this apparent change of notation and/or interpretation of backward messages?
3. In Eq. 7, the partial derivative wrt $\theta$ is computed only for the block local parameters right? But in the true formulation, gradients should be computed for all parameters in the computational graph. Is that correct? Or does the formulation of the probalistic graph enable inferring the true gradients by just computing the block-local parameter gradients? The variational local loss also probably plays a role here. Could you kindly clarify this part as it seems central to the entire proposal?

---

> ### Author Rebuttal · Authors · 2023-08-09
>
>
> We thank the reviewer for ecognising the novelty of the proposed twin-network architecture, and pointing out several ways to improve the paper. We have made a number of changes to make the paper more accessible as suggested, which we will detail below.
>
> We also would like to thank all the reviewers for their constructive comments and questions. Please note that we have uploaded an updated version of our main text as well as supplement. We have indicated all major changes using blue color text. We have also responded to each reviewer separately and in detail.
>
> **The updates are summarised as follows:**
>
> 1. Updates to the notation and theoretical description of the model
> 2. Updated results across the board in Table 1. Large change in performance for CIFAR-10 due to a discovered bug that was affecting just those experiments that caused overfitting. The bug affected the construction of splits in the forward and backward networks to create blocks and local losses. It resulted in effectively only generating meaningful training signals in the last block as suspected by reviewer 5.
> 3. Additional explanatory test and t-SNE results in the supplement.
>
> **Questions:**
> 1) In Like 115 (top of page 4), there seems to be a missing ...
>
> *Response:* Yes, correct. The log in line 115 is a typo. We fixed that, thank you! However, Eq.2 is correct. The identity in Eq.2 is commonly exploited in the EM literature. The trick is to take the derivative on the left side to get 1/p(y|x) and then absorb this term into the expectation to get p(z|x,y). We included a step-by-step expatiation in Sec. S1.4 in the supplement for completeness.
>
> 2) In Eq. 4, you describe to be the backward messages and ...
>
> *Response:* The model uses the forward and backward messages a_k and b_k that correspond to the distributions (3) and (4). The variational posteriors \rho_k are formed by combining a_k and b_k locally. Eq.(7) is the form for a general posterior and Eq.(8) the construction for the specific choice of a_k and b_k. Many of these details were hidden in the supplement, and we re-structured the paper to make this clear.
>
> 3) In Eq. 7, the partial derivative wrt is computed only ...
>
> *Response:* Correct, the proposed method is an approximation to the true gradient. We described this in greater detail in section S1.3.4 where we provide more details to posterior mixing. We also moved more details to the main text to make clear what assumptions were made to arrive at the result in Eq.7.

---

> > ### Comment · Reviewer_HQhi · 2023-08-21
> > **Response to authors**
> >
> > Thank you for the clarification and updating the manuscript based on the reviews.
> >
> > > The log in line 115 is a typo. We fixed that, thank you! However, Eq.2 is correct. The identity in Eq.2 is commonly exploited in the EM literature.
> >
> > Thank you for this clarification. If I understand correctly from the supplementary material, you used $p(y|x) = \mathbf{E}_z [p(y|z_k) P(z_k|x)]$ to get to Eq. 2. I think it would be clearer to write this definition in Line 115 (now Line 121-122).
> >
> > > The model uses the forward and backward messages a_k and b_k that correspond to the distributions (3) and (4).
> >
> > I think I understand this point a bit better now. It was very difficult for me to understand this from the original writing. The updated manuscript probably is a better way of presenting this. However, since the NeurIPS instructions were to upload a 1-page rebuttal only, I haven't been able to get through the updated paper and supplementary closely.
> >
> > > Correct, the proposed method is an approximation to the true gradient.
> >
> > Thanks for this clarification. I must admit this was a misunderstanding on my part, wherein I believed that you had somehow computed the true gradient (with some variance in estimation). But I now understand that this is a biased estimate of the gradient (along with the variance in gradient estimation).
> >
> > Furthermore, based on the comments (+discussion) of reviewer xCFC, I believe that there are some issues in the scalability of this method, specifically around the Cifar-10 training numbers being low. Taken together, I believe that the proposed approach is interesting but probably requires more work/tweaks to be considered as a truly bio-plausible learning rule. Nevertheless, this work could be interesting to the wider NeurIPS community and could serve as a precursor to other bio-plausible learning rules. Therefore, I have readjusted my score.

---

### Official Review · Reviewer_39aM · 2023-07-09

**Soundness:** 3 good
**Presentation:** 3 good
**Contribution:** 4 excellent
**Rating:** 7
**Confidence:** 4

**Summary:**

This paper introduces a novel framework for block-local training of deep networks. It proposes a twin network design that propagates information backwards from targets to the input to provide auxiliary local losses. This design allows forward and backward propagation to occur in parallel preventing the problems of weight transport and locking across blocks. This design is applied to training ResNets and transformers on several tasks.

**Strengths:**

- Overall this paper is clearly written and proposes a novel interesting idea. I think this is an exciting direction that others will be able to build upon.
- The proposed twin architecture and the treatment of block outputs as uncertainty is novel
- The empirical results are strong and show a clear advantage of the block-local learning method particularly on CIFAR-10

**Weaknesses:**

- One selling point of this work is improved training efficiency. I would like to see an analysis even theoretically of what type of speedups can achieved with the proposed block-local training method.
- Although a few different architectures were evaluated the effect of block size on performance was not discussed. This seems like a important parameter to address as it effects both how parallelized/distributed training can be and biological plausibility.

**Questions:**

How would this method scale with block size?

What sort of practical performance speedups can be expected from using your block-local learning? (for example on a system with 1 gpu vs on a system with several gpus)


**Limitations:**

I think the limits of the biological plausibility of the twin architecture where the backwards network requires the same number of parameters as the forwards network should be discussed more.

---

> ### Author Rebuttal · Authors · 2023-08-09
>
> We thank the reviewer for their positive review and recognising the novelty of the twin-network architecture, and the strength of the empirical results. We have added an analysis of the speedup achievable with our model and results related to block-size.
>
> **Responses to specific questions:**
>
> 1) How would this method scale with block size?
>
> *Response:* In the transformer example in Fig.3 we investigated the impact of number of blocks and found that the model seems to scale quite well. Meanwhile we also ran initial experiments for ResNet and also found quite promising results (shown in supplementary section S.2.2.1 and corresponding figure).
>
> 2) What sort of practical performance speedups can be expected from
> using your block-local learning? ...
>
> *Response:* Thank you for this suggestion. In our model inference can be started in both the forward and backward paths in parallel which will lead to a speed up both on single GPU (with sufficient memory) as well as on multiple GPUs. We included pseudo-code and a detailed description of the expected level parallelization compared to backprop in the updated version in Supplementary section S2.

---

### Official Review · Reviewer_3PEq · 2023-07-25

**Soundness:** 3 good
**Presentation:** 2 fair
**Contribution:** 2 fair
**Rating:** 5
**Confidence:** 5

**Summary:**

In this work, the authors address the problem of weight transport and weight locking issue in backprop by introducing a new bio-plausible algorithm known as block-learning to train NNs. The model uses different forward and backward weights, creating a twin network-like scheme to learn efficient signals via local losses. The proposed learning algorithm is tested on convolution and transformer-based architectures to show that the proposed framework can scale to complex architectures.

**Strengths:**

1. Well-written paper
2. Experiments on convolution and transformers show that the proposed work can scale to complex architectures.

**Weaknesses:**

1. Novelty is limited, given current framework shares several similarities with other approaches, such as Local representation alignment.
2. Related work is missing several key citations.
3. Experimental setup is restricted, as several SoTa bio-plausible approaches, including PC based approaches, are not compared

**Questions:**

Other examples of Block-learning framework is Local representation alignment (LRA-E[4], Rec-LRA [5]) , Difference target Propagation (DTP [2], DTP-sigma [4], DTP with backward targets [1], DTP with fixed weights [3]), weight mirroring [11] and Neural Generative Coding (Conv-NGC [6], NGC [8], Act-NGC [7]). The authors should compare and contrast against these existing lines of works, given all these frameworks have shown scaling results on various domains and architectures.

Second several works on predictive coding is not cited [9,10], given they have shown to approximate BP and have shown to achieve similar performance on various benchmarks. Can authors compare against PC based approaches? FA is known to struggle on complex architectures, hence to have

Backprop is known to struggle whenever Bias is set to high value, it is not clear, whether experiments with FA and BP are performed with this settings. If yes, then how does model perform when you set biases to some low number such as 0.01? Can authors report these numbers?


What is the benefit of current approach? Do you observe faster convergence? Better features (for instance one can use T-sne plots to visualize separations between classes, or visualize features of intermediate conv layers)? It would be beneficial if authors can report advantages of proposed approach.

As report by Lillicrap and [4], can authors show update angle compared to BP for the proposed framework? Does model update lies within 90 degree or even 45 degree compared to BP? Such analysis would further strengthen proposed work.

How robust is the model? Can authors report model performance across various settings/hyperparameter settings?

1.	Ernoult, M.M., Normandin, F., Moudgil, A., Spinney, S., Belilovsky, E., Rish, I., Richards, B. and Bengio, Y., 2022, June. Towards scaling difference target propagation by learning backprop targets. In International Conference on Machine Learning (pp. 5968-5987). PMLR.
2.	Lee, D.H., Zhang, S., Fischer, A. and Bengio, Y., 2015. Difference target propagation. In Machine Learning and Knowledge Discovery in Databases: European Conference, ECML PKDD 2015, Porto, Portugal, September 7-11, 2015, Proceedings, Part I 15 (pp. 498-515). Springer International Publishing.
3.	Shibuya, T., Inoue, N., Kawakami, R. and Sato, I., 2023, June. Fixed-Weight Difference Target Propagation. In Proceedings of the AAAI Conference on Artificial Intelligence (Vol. 37, No. 8, pp. 9811-9819).
4.	Ororbia, A.G. and Mali, A., 2019, July. Biologically motivated algorithms for propagating local target representations. In Proceedings of the aaai conference on artificial intelligence (Vol. 33, No. 01, pp. 4651-4658).
5.	https://ojs.aaai.org/index.php/AAAI/article/view/26118
6.	Ororbia, A. and Mali, A., 2022. Convolutional Neural Generative Coding: Scaling Predictive Coding to Natural Images. arXiv preprint arXiv:2211.12047.
7.	Ororbia, A.G. and Mali, A., 2022, June. Backprop-free reinforcement learning with active neural generative coding. In Proceedings of the AAAI Conference on Artificial Intelligence (Vol. 36, No. 1, pp. 29-37).
8.	Ororbia, A. and Kifer, D., 2022. The neural coding framework for learning generative models. Nature communications, 13(1), p.2064.
9.	Millidge, B., Salvatori, T., Song, Y., Bogacz, R. and Lukasiewicz, T., 2022. Predictive coding: towards a future of deep learning beyond backpropagation?. arXiv preprint arXiv:2202.09467.
10.	Salvatori, T., Pinchetti, L., Millidge, B., Song, Y., Bao, T., Bogacz, R. and Lukasiewicz, T., 2022. Learning on arbitrary graph topologies via predictive coding. Advances in neural information processing systems, 35, pp.38232-38244.
11.	Akrout, M., Wilson, C., Humphreys, P., Lillicrap, T. and Tweed, D.B., 2019. Deep learning without weight transport. Advances in neural information processing systems, 32.

**Limitations:**

1. Comparison is needed with relevant methods.
2. Analysis and ablation study missing.

---

> ### Author Rebuttal · Authors · 2023-08-09
>
> We thank the reviewer for the valuable comments and pointers to previous literature. We disagree that our algorithm is limited in novelty — in fact there are several novel and key differences between our work and the works referred by the reviewer as described below. Since the primary goal of our approach is more scalable training, the PC are a bit orthogonal. We have included references to them and a result comparison nevertheless.
>
> **Responses to questions:**
>
> 1) Other  examples of Block-learning framework is Local representation alignment ...
>
> *Response:* We included these references and also the suggested T-SNE analysis in the updated version (see Figure S1 in the supplement). Note that Ororbia et al. 2023 was published after the NeurIPS 2023 submission deadline and therefore we couldn’t know about this result. The PC literature is a bit orthogonal but gives a very nice additional view on the broader topic of block-local learning. We believe that the probabilistic formulation of the framework that comes with explicit per-block uncertainty estimates are conceptually novel and beneficial for distributed training of the model compared to previous methods. We further discussed this and provided additional analysis. We added additional baselines as suggested. See updated versions of the related work section 2, updated table 1 with additional benchmarks and updated experimental results.
>
> 2) s report by Lillicrap and [4], can authors show update angle ...
>
> *Response:* Thank you for this suggestion. We made a first test based on the ResNet-18 experiments and found that in fact the angles were consistently below 90 degrees but rarely below 45. We will include a detailed analysis in the camera ready version of the paper.
>
> 3) How robust is the model? Can ...
>
> We added first additional experiments with varying number of blocks (and thus different blocks sizes). The results look quite promising, the network seems to scale well with the number of blocks. We included this analysis in the updated version of the paper and will add additional ablation studies also for other tasks for the camera ready version.

---

### Official Review · Reviewer_RmC3 · 2023-07-26

**Soundness:** 2 fair
**Presentation:** 2 fair
**Contribution:** 3 good
**Rating:** 5
**Confidence:** 2

**Summary:**

The authors propose a novel approach to the estimation of deep neural network parameters using block-localized backpropagation in conjunction with belief propagation. This approach is much more parallelizable, thus should help for distributed training, enabling horizontal scaling across devices.

The proposed approach works using a twin *backward* network, and by incorporating the belief messages into the block-local losses which are optimized using gradient descent.


**Strengths:**

- The proposed approach is an interesting combination between belief propagation and back-propagation. I particularly appreciated the view of a neural network as a Markov chain.
- The proposed approach is significant, especially considering the current state of deep learning research. Large neural networks are steadily becoming the norm, and the development of specific learning algorithms for this kind of models is a valid and important research direction.

**Weaknesses:**

- The paper is sometimes not clear. I personally had difficulties understanding the following sections;
    - 3.1, especially after equation (2)
    - 3.4, it seems it requires the supplementary material to be correctly comprehended.
- I think with the current state of the paper it might be difficult to reproduce the reported results. The algorithm itself is not completely clear, and I think the submission would improve from a clear explanation (maybe provided in the supplementary material).
- A more in-depth analysis of the BLL algorithm convergence would improve the paper's strength.

**Questions:**

- How big should the twin network be? As much as the forward network, or have you found that it can be smaller?
- This is a suggestion, but I think the submission will benefit from a clear description of the algorithm (possibly written in pseudo-code). This can be appended in the supplementary material and then referenced in the main paper.
- Can a Gaussian distribution with non-constant variance be used to model the layer probabilities, or would the algorithm not work with this assumption?
- I believe there is an error in the equations S9, S10, S11, S13. I imagine index *k* should not be one of the variables of the summation.
- I believe there is an error in line [115]. I imagine it should be *p(y|x)* instead of *log p(y|x)*.
- This is a simple suggestion, but in my experience (in the ML community) it seems to be more common to use 𝔼[…] and ∇ to indicate the expected value and gradient respectively. For a more standardized notation, I personally would favor that nomenclature. If you think that would be detrimental to the paper presentation in any way, you are free to ignore this suggestion.

**Limitations:**


- The algorithm requires to train an additional twin network, thus the number of total parameters optimized is greater than with standard gradient descent.
- The message-passing operation could affect convergence time, Fig. 3 seems to suggest otherwise, but it may not be always holtrue for other dataset or hyper-parameters.

---

> ### Author Rebuttal · Authors · 2023-08-09
>
> The authors would like to thank the reviewer for the valuable comments and recognising the novelty and significance of the approach. We have improved clarity of the paper and added more details about the algorithm and hyper-parameters.
>
> **Further responses inline:**
>
> 1) How big should the twin network be? As much as the forward network, or have you found that it can be smaller?
>
> *Response:* The transformer model in the submitted version already had a simple backward structure that was not as large as the forward network, which still worked well. We will include further experiments with the resnet architectures as well to study this issue further.
>
> 2) This is a suggestion, but I think the submission will benefit from a ...
>
> *Response:* We added pseudo-code to Supplement section S2
>
> 3) Can a Gaussian distribution with non-constant variance be used ...
>
> *Response:* Yes, that is also possible. Multi-parameter exponential family distributions can be easily accommodated in the theoretical framework by splitting the network outputs into multiple parts when computing the local loss. Each part then e.g. represents mean or variance respectively.
>
> 4) I believe there is an error in the equations S9, S10, S11, S13. I imagine index *k* should not be one of the variables of the summation.
>
> *Response:* Yes, the index is used twice here. Thank you for pointing this out. We fixed the notation in the updated version.
>
> 5) I believe there is an error in line [115]. I imagine it should be *p(y|x)* instead of *log p(y|x)*.
>
> *Response:* Correct, we fixed that. Thank you!
>
> 6) This is a simple suggestion, but in my experience (in the ML ...
>
> *Response:* We updated the notation to the more standard version in the updated version.

---

> > ### Comment · Reviewer_RmC3 · 2023-08-19
> >
> > I thank the authors for the response and the changes introduced in the rebuttal. I appreciate the effort the authors put into improving the manuscript (I find that the pseudo-code in Table S4 is particularly useful in my opinion). I however feel that the paper might still need work regarding the overall clarity of the proposed approach, so I have decided to maintain my current rating.

---

### Author Rebuttal · Authors · 2023-08-09

**General response to the reviewers:**

We would like to thank all the reviewers for their constructive comments and questions. Please note that we have uploaded an updated version of our main text as well as supplement. We have indicated all major changes using blue color text. We have also responded to each reviewer separately and in detail.

The updates are summarised as follows:

1. Updates to the notation and theoretical description of the model
2. Updated results across the board in Table 1. Large change in performance for CIFAR-10 due to a discovered bug that was affecting just those experiments that caused overfitting. The bug affected the construction of splits in the forward and backward networks to create blocks and local losses. It resulted in effectively only generating meaningful training signals in the last block as suspected by reviewer 5.
3. Additional explanatory test and t-SNE results in the supplement.

---

### Decision · Program_Chairs · 2023-09-21

**Decision:**

Reject

**Comment:**

This paper presents a block-local learning rule, which, under some distributional assumptions on the network representations, can be understood as a type of message passing between blocks, which in turn can be formulated as a collection of local losses that can be then optimized using SGD, leading to a new learning rule which is studied and investigated empirically.

The reviewers agree that the topic of the paper is important and will be interesting to to the community. The empirical analysis seems fairly comprehensive, and the reviewers appreciated the strong motivation and discussion of the probabilistic interpretation and message passing perspective. Several reviewers felt that the algorithm itself was quite hard to understand, as key ingredients of the argument and analysis were moved to the appendix and were not explained clearly. In particular, the intractable term in the ELBO was mentioned as a source of confusion on several fronts, both from the algorithmic component, as well as a possible impediment to parallelizing the forward and backward pass.

The paper as a whole could benefit from improved clarity of writing, better positioning with respect to prior work, additional comparisons to appropriate baselines, and some structural reorganization. As such, I recommend that the authors thoroughly address all the reviewer concerns and resubmit the manuscript to a future venue.